

# Impacts of Biogenic Emissions on Summertime Ozone Formation in the Guanzhong Basin, China

Nan Li [1,2], Qingyang He [2,3], Jim Greenberg [4], Alex Guenther [5], Junji Cao [2,6], Jun
Wang [7], Hong Liao [1], Qiang Zhang [8]

[1]Jiangsu Key Laboratory of Atmospheric Environment Monitoring and Pollution
Control, Jiangsu Engineering Technology Research Center of Environmental
Cleaning Materials, Jiangsu Collaborative Innovation Center of Atmospheric
Environment and Equipment Technology, School of Environmental Science and
Engineering, Nanjing University of Information Science & Technology, Nanjing
210044, China

[2]Key Lab of Aerosol Chemistry & Physics, SKLLQG, Institute of Earth Environment,
Chinese Academy of Sciences, Xi'an, 710061, China

[3]Nanjing Star-jelly Environmental Consultants Co., Ltd, Nanjing, 210013, China

[4]National Center for Atmospheric Research, CO, USA

[5]Department of Earth System Science, University of California, Irvine, Irvine CA
92697-3100, USA

[6]Institute of Global Environmental Change, Xi'an Jiaotong University, Xi'an, 710049,
China

[7]Center of Global and Regional Environmental Research, Department of Chemical
and Biochemical Engineering, University of Iowa, Iowa City, Iowa, USA

[8]Department of Earth System Science, Tsinghua University, Beijing, 100084, China

*Correspondence to: Nan Li, linan@nuist.edu.cn*



## Abstract

This study is the first attempt to understand the synergistic impact of anthropogenic and biogenic emissions on summertime ozone ($O_3$) formation in the Guanzhong (GZ) basin where Xi'an, the oldest and the most populous city (with a population of 9 million) in the northwest China, is located. Month-long (August 2011) WRF-Chem simulations with different sensitivity experiments were conducted and compared with near-surface measurements. Biogenic volatile organic compounds (VOCs) concentrations were characterized from 6 surface sites among the Qinling Mountains, and urban air composition was measured in the Xi'an city at a tower 100 m above the surface. The WRF-Chem control experiment reasonably reproduced the magnitudes and variations of observed $O_3$, VOCs, $NO_x$, $PM_{2.5}$ and meteorological parameters, with normalized mean biases for each parameter within ±21%. Subsequent analysis employed the factor separation approach (FSA) to quantitatively disentangle the pure and synergistic impacts of anthropogenic and/or biogenic sources on summertime $O_3$ formation. The impact of anthropogenic sources alone was found to be dominant for $O_3$ formation. Although anthropogenic particles reduced $NO_2$ photolysis by up to 60%, the anthropogenic sources contributed 19.1 ppb $O_3$ formation on average for urban Xi'an. The abundant biogenic VOCs from the nearby forests promoted $O_3$ formation in urban areas by interacting with the anthropogenic $NO_x$. The calculated synergistic contribution (from both biogenic and anthropogenic sources) was up to 14.4 ppb in urban Xi'an, peaking in the afternoon. Our study reveals that the synergistic impact of individual source contributions to $O_3$ formation should be considered in the formation of air pollution control strategies, especially for big cities in the vicinity of forests.

**Keywords:** ozone, emission, biogenic volatile organic compounds, WRF-Chem



## 1. Introduction

Elevated ozone ($O_3$) levels in China has drawn increasing attention in recent years (e.g. Xue et al., 2014; Hu et al., 2016; Wang T. et al., 2016; Wang M. et al., 2016). $O_3$, a secondary pollutant, is mainly formed by complex photochemical

reactions of nitrogen oxides ($NO_x = NO + NO_2$) and volatile organic compounds (VOCs). High $O_3$ concentration at ground level is harmful to human health and ecosystems (WHO, 2005; Feng et al., 2015; Brauer et al., 2016). $O_3$ in the troposphere is an important greenhouse gas that has the third-highest radiative forcing after $CO_2$ and $CH_4$ (Stevenson et al., 2013; IPCC, 2013). In addition, $O_3$ is the primary source of

the hydroxyl radical (OH), which has a major influence on the oxidizing capacity of the atmosphere and thus impacts the oxidation chemistry of secondary pollutants (e.g. sulfate and secondary organic aerosol) (Ehhalt et al., 2000; Rohrer et al., 2006). In recent years, the surface $O_3$ level has been increasing in most Chinese cities. For instance, in the highly urbanized areas in China, maximum 8h $O_3$ concentration

increased by 98% (22 ppb) from 2013 to 2017 in the Yangtze River Delta (YRD) region, and the growth trend was 41% (13 ppb) for Beijing-Tianji-Hebei (BTH) region and 30% (10 ppb) for the Pearl River Delta (PRD) region (http://datacenter.mep.gov.cn). In the future, the pollution trend is likely to worsen due to potential changes of climate and emissions (Wang et al., 2013; Liu et al., 2013;

Zhu et al., 2016, 2017).

On the global scale, VOC emissions from natural vegetation is estimated to be one order greater than that from anthropogenic activities, in spite of large uncertainties in different studies (Guenther et al., 2006; Wu et al., 2007, 2008; Jiang et al., 2013; Zhu et al., 2017). In addition, biogenic VOCs (e.g. isoprene) are highly

reactive, reacting more efficiently with OH than most anthropogenic VOC species (Carlo et al., 2004). Previous studies have demonstrated the significant impacts of biogenic VOCs on surface $O_3$ formation under strong solar radiation, high temperature and $NO_x$ level (e.g. Fiore et al., 2005, 2011; Wang et al., 2008; Curci et al., 2009; Geng et al., 2011; Strong et al., 2013; Squire et al., 2014; Lee et al., 2014;

Zhang et al., 2017). In China, biogenic VOC emissions are estimated to be 17-44 TgC $yr^{-1}$ (Guenther et al., 1995; Lin et al., 2008; Fu et al., 2012, 2104; Li et al., 2014), and are concentrated in the warm summer season. Current studies report that biogenic VOCs contribute to surface $O_3$ concentrations in China (Geng et al., 2011; Qu et al.,



2013, 2014) and play an important role in intercontinental transport of $O_3$ (Zhu et al., 2017).

The impacts of biogenic VOCs on $O_3$ formation may vary in different regions and different seasons (Im et al., 2011; Strong et al., 2013; Wagner et al., 2014; Lee et al., 2014). Qu et al. (2013) employed the RAQM model to examine the influence of biogenic emissions on daily maximum surface $O_3$ concentration in China. Their calculations showed that in general the impact from biogenic sources on $O_3$ were more obvious in South China than in North China, but the $O_3$ increments in different regions didn't follow the same seasonality. Geng et al. (2011) used the WRF-Chem model to evaluate the effect of biogenic emissions on $O_3$ production in Shanghai in summer, and suggested that the carbonyls produced by the continuous oxidation of isoprene have important impacts on $O_3$ level in the city. In addition, some studies suggest that biogenic emissions may increase due to global warming and land-use change, and the impact on $O_3$ formation could be more significant in the future (Lin et al., 2008; Fiore et al., 2011; Liu et al., 2013; Fu et al; 2015). Fiore et al. (2011) pointed out that the potential increases in biogenic isoprene in North American (NA) could offset the regional and intercontinental surface $O_3$ decreases produced by controls on NA anthropogenic emissions during warm seasons.

The Guanzhong (GZ) basin is the most developed region in northwestern China. In the past few years, air pollution has grown up to be a severe issue in the GZ basin (Wang et al., 2012; Xue et al., 2017), due to its specific basin topography and abundant anthropogenic emissions (Li et al., 2017). According to the data from national environmental monitoring stations in the GZ basin, 43% of days in 2013-2017 have AQI > 100 (e.g., unhealthy air quality category), and in summer $O_3$ was regarded as the primary pollutant in 70% of polluted cases. In this study, we employed a regional chemical model WRF-Chem to simulate $O_3$ concentration in the GZ basin for summer 2011. Our aim is the source apportionment of urban $O_3$ formation in this city surrounded by forests, specifically quantifying the individual and synergistic contributions of anthropogenic and biogenic sources. The paper is organized as follows. We first describe the sampling campaign, the chemical model and the emission data used for driving the model (Section 2). We then evaluate the model performance by comparing the observed urban air quality and biogenic VOCs



with the simulated results (Section 3). Finally, we analyze the sensitivity of summertime O₃ formation to biogenic and anthropogenic sources (Section 4).

## 2. Methodology

### 2.1 Sampling sites and descriptions

The study was conducted in Xi'an, one of the oldest cities in the world. Xi'an is the most populous city in the northwest of China with a population of greater than 9 million. It is located in the heartland of the GZ basin between the Qinling Mountains and the Loess Plateau (Fig. 1). The topographical features result in air pollutants often being trapped in the valley with limited dispersion. The city borders the northern foot of the Qinling Mountains, only around 50 kilometers from downtown to the foothills. The Qinling Mountains are an east-west mountain range of 1600 kilometers in length and 300 kilometers in width and are regarded as a natural boundary between Northern and Southern China. Climate, and even culture are significantly different from the north to the south. The Xi'an city has high temperatures and strong solar intensity in summer, making it an ideal location to assess the importance of biogenic contributions to urban air quality.

2.1.1 Biogenic VOC measurements in the Qinling Mountains

We collected ambient air samples at six field sites in the Qinling Mountains (Fig. 1b, the triangles) on 6th – 7th August 2011 under sunny weather conditions (details are presented in Table 1). Sampling was conducted between 9:30 am to 16:30 pm local time to target expected daily maximum isoprene concentrations. At each site, ambient air samples were pulled in parallel onto three cartridges filled with Tenax GR and Carbograph 5TD solid adsorbents using a mass flow controlled pump for 30 min. Samples were shipped to the lab at NCAR (Boulder CO, USA) for chemical analysis. Cartridges were desorbed using an Ultra™ TD auto sampler with a Unity thermal desorption system (MARKES International Series 2, Llantrisant, UK) interfaced with a temperature programmed Agilent 7890A series Gas Chromatograph with a 5975C Electron Impact Mass Spectrometer and a Flame Ionization Detector (GC-MS/FID, Agilent Technologies, Santa Clara, CA, USA). We used nitrogen as a carrier gas at the flow rate of 3 mL min⁻¹. Isoprene and monoterpene identifications were based on



the comparison of retention time of authentic standards and mass spectra in the National Institute of Standards and Technologies (NIST) databases. Quantifications were calculated using FID calibrated with a NIST traceable standard.

2.1.2 Air quality monitoring in urban Xi'an

We set up an urban air quality monitoring site (Fig. 1b) at the roof (107 m above ground) of the main building (108.984 °E, 34.245 °N) on the campus of Xi'an Jiaotong University to minimize the ground level influences of local emissions. The campus is in the southeastern part of the downtown surrounded by residential areas. We obtained reliable observations of the concentrations of $O_3$, $NO_x$ and $PM_{2.5}$ during $15^{th}$ – $30^{th}$

August 2011. Gases were measured by Ecotech analyzers (Ecotech Pty Ltd, Australia). $O_3$ were measured by an UV photometric analyzer EC9801. $NO_x$ was measured by a gas-phase chemiluminescence detection analyzer EC9841. We collected 24h $PM_{2.5}$ filter samples by a mini-volume sampler (Airmetrics, USA) at a flow rate of 5 L $min^{-1}$ using both 47 mm quartz-fiber (Whatman, Mid-dlesex, UK) and teflon-membrane

(Gelman, Ann Arbor, MI) filters. We calculated the $PM_{2.5}$ mass concentrations gravimetrically by weighing the teflon-membrane filters pre and post-collection for at least 4 times using an electronic microbalance (MC5, Sartorius, Göttingen, Germany) with ±1 μg sensitivity under controlled conditions. EC and OC concentrations were analyzed based on a 0.5 $cm^2$ punch from the quartz-fiber filter following the

IMPROVE_A (Interagency Monitoring of Protected Visual Environments) thermal/optical reflectance (TOR) protocol (Chow et al., 2007) using a DRI model 2001 Carbon Analyzer. The concentrations of ions were quantified from a 10.8 $cm^2$ of the teflon-membrane filter by a Dionex DX-600 ion chromatography (Dionex Inc., Sunnyvale, CA, USA) (Zhang et al., 2014).

**2.2 The WRF-Chem model**

    We employed the WRF-Chem model to study biogenic VOC emissions from the Qinling Mountains and their contributions to regional $O_3$ formation in urban Xi'an. WRF-Chem is a 3-D online-coupled meteorology and chemistry model consisting of

the components of emission, transport, chemical transformation, photolysis and radiation (Tie et al., 2003, Li et al., 2011), dry and wet deposition (Wesely, 1989), and aerosol interactions (replaced with CMAQ aerosol module, Binkowski and Roselle,



2003; Li et al., 2010). WRF is a non-hydrostatic mesoscale dynamical system with various options for physical parameterizations (Skamarock et al., 2008). The chemical modules were implemented into the WRF framework obeying the same schemes for the simultaneous simulations (Grell et al., 2005).

We adopted RADM2 (Regional Acid Deposition Model) as the gas phase chemical mechanism to predict $O_3$ formation. RADM2 is an aggregated species type using the reactivity based weighting scheme to adjust for lumping (Stockwell et al., 1990). The mechanism implemented in our WRF-Chem model covers 158 reactions among 36 species, containing the complete reaction paths for isoprene, monoterpenes
and the relevant inorganic reactions. As an explicit species, isoprene chemistry is based on an updated CB4 gas-phase mechanism (Carter and Atkinson., 1996). Monoterpenes are represented by a unique species TERP, with the reactions of OH, $O_3$, and $NO_3$. These reactions use the same rate constants as the reaction of OLI (Internal olefins), and have the TERPAER counter species added to track the
throughput of this reaction.

     The simulated domain (Fig. 1) is 600×600 $km^2$ centered on urban Xi'an with 3 km horizontal grid spacing. We set up 28 vertical layers from the surface up to 50 hPa with 7 layers below 1 km to assure a high near-ground vertical resolution. The National Centers for Environmental Prediction (NCEP) FNL Operational Global
Analysis data provided the initial and boundary fields of meteorology. Initial and boundary conditions of chemistry were derived by a global chemical transport model (Model for Ozone and Related chemical Tracers, MOZART) (Emmons et al., 2010). We considered the first 7 days as spin-up period, and the study focused on $6^{th} – 7^{th}$ and $15^{th} – 30^{th}$ August 2011 because of the available field observation datasets (as
described in Section 2.1).

### 2.3 Biogenic and anthropogenic emissions

     Biogenic emissions were quantified by the widely used model MEGAN (Model of Emissions of Gas and Aerosols from Nature) (Guenther et al., 2006). MEGAN
coupled into the WRF-Chem model, referred to here as WRF-MEGAN2, provides on-line estimates of the net landscape-averaged biogenic emissions from terrestrial ecosystems into the above-canopy atmosphere. The on-line estimated emissions of




isoprene, individual monoterpenes and other biogenic VOCs serve as the inputs for the further chemistry simulation. To drive MEGAN, we need the following inputs: emission factors (EFs), leaf area index (LAI), plant functional types (PFTs), as well as meteorology conditions. The meteorology was obtained from WRF simulations and

5 the LAI and PFT data were extracted from MODIS (Tian et al., 2004). We adopted the canopy-scale emission factors of dominant species from Guenther et al. (2006).

Estimated the whole year 2011 by WRF-MEGAN2, the isoprene and monoterpene emissions in the Qinling Mountains were mostly concentrated in summer (71% and 58%, respectively). During our simulation period, the isoprene emission from the

10 domain is 157 Gg mon$^{-1}$, accounting for ~80% of total biogenic VOC emissions (Table 2). The rest are monoterpenes and other biogenic VOCs (e.g. acetone and MBO (2-methyl-3-buten-2-ol)). Figures 2a and 2b show the spatial distributions of biogenic isoprene and monoterpenes emission fluxes during the simulation period, indicating the high emission zone of isoprene in the Qinling Mountains lying to the

15 south of the Xi'an city.

The anthropogenic emissions were obtained from the Multi-resolution Emission Inventory for China (MEIC, Li et al., 2017) for the year of 2010, which was downscaled to a resolution of 3 km using locations of point sources and various spatial proxies (Geng et al., 2017). The upgraded highly resolved emission data were

20 based on a collection of statistics and newly developed emission factors. The emission inventory used in our model includes all major anthropogenic sources, but excluded open biomass burning which occupies a low proportion in the GZ basin during our simulating period (estimated by Fire Inventory from NCAR, https://www2.acom.ucar.edu/modeling/finn-fire-inventory-ncar). The anthropogenic

emission sources are composed of power, industry, residential, transportation, and agriculture. The inventory of VOCs, SO$_2$, NO$_x$, NH$_3$, and PM$_{2.5}$ in the domain during the simulation period is summarized in Table 2. The estimated anthropogenic VOCs emissions are 72.2 Gg, contributing up to ~30% of total VOC emissions. Figures 2c and 2d present the spatial distributions of anthropogenic VOC and NO$_x$ emissions in

the simulation period. The highest emission intensity of anthropogenic VOCs and NO$_x$ are in Xi'an city and the GZ basin due to the frequent vehicle and industrial activities of this area.



### 2.4 Factor Separation Technique

O$_3$ is formed by complicated nonlinear reactions of anthropogenic and biogenic precursors (NO$_x$ and VOCs) in the presence of sunlight. The approach referred to as the "brute-force" method is traditionally used in air quality model to identify source contributions from specific non-reactive species in a linear process, but it is not straightforward to quantify the contributions of several different factors in a nonlinear process. In practice, the actual impact of one factor in a nonlinear process in the presence of others can be separated into 1) pure impact from the factor and 2) interactional impacts from all those factors. In this study, we adopted the factor separation approach (FSA) (Stein and Alpert, 1993) to decompose the pure contribution of a factor from its interaction with other factors.

We considered anthropogenic and biogenic sources as two interactional factors to influence the O$_3$ formation. $f_{anth\text{-}bio}$, $f_{anth}$, $f_{bio}$ and $f_0$ are the simulation results including both anthropogenic and biogenic sources, anthropogenic source only, biogenic source only, and neither, respectively. Pure contributions of anthropogenic and biogenic sources are expressed as Eq. (1) and (2), respectively:

$$f'_{anth} = f_{anth} - f_0 \qquad (1)$$
$$f'_{bio} = f_{bio} - f_0 \qquad (2)$$

The calculated result including both anthropogenic and biogenic sources should include both pure contributions of the two factors, the synergistic impact, and the impact of background transport (Eq. 3):

$$f_{anth-bio} = f'_{anth} + f'_{bio} + f'_{anth-bio} + f_0 \qquad (3)$$

Thus, the synergistic effect between anthropogenic and biogenic sources is represented as:

$$
\begin{aligned}
f'_{anth-bio} &= f_{anth-bio} - f'_{anth} - f'_{bio} - f_0 \\
&= f_{anth-bio} - (f_{anth} - f_0) - (f_{bio} - f_0) - f_0 \qquad (4) \\
&= f_{anth-bio} - f_{anth} - f_{bio} + f_0
\end{aligned}
$$

Based on the FSA, we conducted four simulations, namely BASE, ANTH, BIO and NEITHER, to explore the pure and synergistic impacts of anthropogenic and/or





biogenic sources on $O_3$ production in the GZ basin. Detailed simulation settings and the various contribution definitions are summarized in Table 3.

### 3. Observation data and model validation

**3.1 Meteorology**

The specific topographical features of Xi'an make the meteorological conditions crucial for the accumulation and dispersion of urban pollutants. To validate our model performance in wind, temperature and relative humidity, we compared the hourly meteorological data (http://www.meteomanz.com) observed at the Jinghe site

(108.58 °E, 34.26 °N, in the west of Xi'an) with model simulations. Figure 3c illustrates the observed and simulated near-surface wind speed and directions during $15^{th} - 30^{th}$ August 2011. The WRF-Chem model successfully captured the prevailing wind direction from north and northeast, consistent with the in-situ observations. It should be noted that in our simulation period the prevailing wind blew from south,

which enhanced the transport of biogenic emissions from the Qinling Mountains to urban Xi'an. In addition, a continuous rainfall event during $18^{th}$-$22^{nd}$ August (green shadow in Fig. 3) was characterized by lower temperature and near-saturated humidity.

We conducted the statistical verification of meteorological variables in Table 4,

including the r (correlation coefficient), NMB (normalized mean bias) and RMSE (root mean square errors). Modeled meteorological variables were in good agreement with observations (Fig. 3a-c) with the NMB less than ±6%.

**3.2 Biogenic VOC concentrations in the Qinling Mountains**

Samples from the Qinling Mountains show that the dominant VOC species was isoprene, and α–pinene was the main constituent of monoterpenes (Table 1). The ratio of isoprene to monoterpenes varies considerably. We compared the VOC measurements with model simulations to validate whether the calculated results were reasonable. The isoprene mean concentration simulated in the six grids was 1.4 ppb,

which is close to the observed average value of 1.3 ppb at the six sampling sites.



Monoterpenes performed quite similarly, simulated 0.22 ppb comparing with observed 0.21 ppb. The evaluation indicates that biogenic VOCs simulations reasonably agreed with the observations in the Qinling Mountains, on average, which provides a basis for us to further evaluate biogenic effects on $O_3$.

### 3.3 Gaseous and particulate pollutants in urban Xi'an

The sampling campaign was organized in summer. Based on the gaseous and particulate pollutant observations, the daily mean $PM_{2.5}$ concentration was $90.0\pm53.5$ $\mu g\ m^{-3}$, with 57% of days exceeding the WHO Interim target-1 (IT-1) 75 $\mu g\ m^{-3}$. The

10 daily mean $NO_x$ concentrations were 25.8-63.2 ppb, with 40% of days exceeding the guideline 48.7 ppb ($\approx$100 $\mu g\ m^{-3}$, GB 3095-2012). The maximum 8h $O_3$ concentration was 3.5-95.6 ppb, with most of the values around the national first grade standard of 46.6 ppb ($\approx$100 $\mu g\ m^{-3}$, GB 3095-2012). Summer in Xi'an is monitored as the least polluted season of the year, and the case we picked is regarded as a typical situation in

summer Xi'an.

Figure 3 compares the simulated hourly $O_3$ and $NO_x$ concentrations with in-situ observations. During the rainy episode (the green shadow in Fig. 3), our model overestimated $NO_x$ concentration and underestimated $O_3$ concentration. The deviation can be explained by the failure to simulate the precipitation in the WRF model

resulting in underestimates in wet deposition. So, we focused our analysis on the period excluding the rainy period. During the no-raining days, our model well reproduced the diurnal variations and magnitudes of $O_3$ and $NO_x$ concentrations. The simulated hourly $NO_x$ averaged for the no-raining period was 46.6 ppb, close to the observed 47.0 ppb (NMB=-1%). For $O_3$, the calculated result averaged for no-raining

period was 38.7 ppb, ~20% higher than the observed value of 31.5 ppb. Our simulated $O_3$ also reproduced the temporal variation of the observed $O_3$ (r = 0.72).

We analyzed $PM_{2.5}$ concentration and composition (sulfate, nitrate, ammonium, EC, organic matter) with the filter-base measurements. The simulated $PM_{2.5}$ concentration was $94.6\pm28.2$ $\mu g/m^3$, slightly lower (NMB=-12%) than measured 107

$\mu g/m^3$ averaged for the no-raining period. The simulated results agreed well with observed $PM_{2.5}$ compositions (Fig. 4b and 4c). Sulfate is the dominant constituent of both simulated (32%) and observed (37%) $PM_{2.5}$. High sulfate concentration was



mainly attributed to the high $SO_2$ emission in the GZ basin as well as the humid weather conditions (Wang et al., 2014). The secondary constituent of observed $PM_{2.5}$ is organic matter, which accounted for 16% of the total observed $PM_{2.5}$, close to the simulated result (14%). Secondary organic matter contributed half to total simulated

organic matter, mainly due to the abundant precursor (i.e. VOCs) emissions and the high atmospheric oxidation capacity in summer.

## 4. Impacts of anthropogenic and biogenic sources on $O_3$ formation

In this section, we analyzed the results from the four simulations (BASE, ANTH,

BIO and NEITHER, Table 3) to characterize the fate of $O_3$ and its precursors in the GZ basin and to quantify the pure and synergistic impacts of anthropogenic and/or biogenic sources on summertime $O_3$ formation.

### 4.1 Base simulation of $O_3$

Firstly, we discussed the spatial and temporal characteristics of the simulated $O_3$

and the precursors (VOCs and $NO_x$) in the GZ basin in the BASE simulation.

Figure 5a shows spatial distribution of the simulated VOCs during the no-raining period, overlaid with the simulated wind vectors. The highest concentration (more than 50 ppb) was in urban Xi'an and its downwind region (the southwest of urban Xi'an), due to anthropogenic activities. In addition, another high-value area (~30 ppb)

was found in the Qinling Mountains, which was probably due to biogenic sources. To better understand the composition of VOCs, we analyzed some typical individual VOC species. Figure S1a shows the spatial distribution of xylenes, representing anthropogenic VOCs, and Fig. S2a and S3a show isoprene and monoterpenes, representing biogenic VOCs. The anthropogenic xylenes were mainly distributed in

the GZ basin, while the high biogenic isoprene and monoterpene concentrations were found over the Qinling Mountains. These results explain the spatial feature of total VOCs and the dominant sources. Detailed discussion of source apportionment is given in Section 4.2

The spatial distribution of $NO_x$ was slightly different (Fig. 6a). The highest

concentrations of $NO_x$ were in the GZ basin (average of 11.1 ppb), especially in urban





Xi'an (averaged of 30.1 ppb), while among the Qinling Mountains, $NO_x$ was low and dominated by biogenic sources.

PM, even though not directly involved in the formation pathways of $O_3$, influences the chemical equilibrium indirectly. In the daytime, $NO_2$ photolysis

frequency ($J(NO_2)$) is determined by the solar radiation influenced by PM via scattering and absorption. Figure 7 shows the changes of $J(NO_2)$ under the participation of PM averaged for urban Xi'an. $J(NO_2)$ was reduced by 40-60%, most significantly in morning and evening rush hours. In the night time, $PM_{2.5}$ can remove $N_2O_5$ from the $NO_x$ cycle via heterogeneous reactions, as one of the major $NO_x$ sinks

in the atmosphere (Xue et al., 2014). Figure 8a shows the spatial feature of $PM_{2.5}$. The densest area was urban Xi'an (averaged for 102 $\mu g\ m^{-3}$) followed by the western part of the GZ basin. The spatial distribution of high-values of $PM_{2.5}$ was similar to that of $NO_x$, but covered a wider area mostly in the downwind region of urban Xi'an, which is expected due to the time required for secondary aerosol formation and thus further

dispersion.

The typical diurnal variation of $O_3$ (Fig. 9b) demonstrates there are higher concentrations in the afternoon and lower at night. For better understanding of $O_3$ concentration characteristics and source/sink mechanisms, we discussed two different time scales: 1) $O_3$ peak time (14:00-18:00) (Fig. 10) and 2) $O_3$ 24-hour average (Fig.

11). During the peak time, simulated near-surface $O_3$ was high in the GZ basin, with averaged concentration of 75 ppb. In the downwind region of high $NO_x$ and VOCs in the west of urban Xi'an, the concentration reached up to 110 ppb. We employed the ratio of $H_2O_2/HNO_3$ to investigate the chemistry regime of $O_3$ formation (Sillman 1995; Wang et al., 2017). If the ratio is greater than 0.5, the $O_3$ production regime is

considered $NO_x$-controlled, otherwise VOC-controlled if the ratio less than 0.3. The range between 0.3 and 0.5 is defined as the transition regime from $NO_x$- to VOC-controlled, indicating the competition of both $NO_x$ and VOCs in $O_3$ production. Figure 12a shows the spatial distribution of the simulated $H_2O_2/HNO_3$ ratio during the $O_3$ peak time. The urban Xi'an and the west and southeast of the GZ basin were right

in the transition regime with a complicated $O_3$ production mechanism sensitive to both $NO_x$ and VOCs. Most of the rest of the simulation region was VOC-controlled, excluding the Yuncheng and Hejing cities in the neighboring Shanxi provinces.



On the 24h average scale, the spatial distribution of $O_3$ presented a different picture (Fig. 11). The original high-value area during the peak time in the GZ city cluster shifted to low-value region due to the consumption of $O_3$ by abundant $NO_x$ emissions. At night time, the titration effect of freshly emitted NO dominates,and the

$O_3$ concentration tends to drop to a lower level. The high value of 24h averaged $O_3$ converged in the south and northwest outside of the GZ Basin. Those areas have elevated $O_3$ due to high daytime production, similar to the nearby zone of peak $O_3$, but also have lower emissions of NO resulting in lower loss of $O_3$.

**4.2 Pure impact of anthropogenic or biogenic sources**

Using the FSA method, we evaluated the pure contribution of anthropogenic or biogenic sources to the summertime $O_3$ formation in the GZ basin. In the scenario of pure contribution of anthropogenic emissions, the VOC concentrations were mostly distributed over the GZ city cluster (8.0 ppb), especially in urban Xi'an (26.4 ppb)

(Fig. 5c). In the scenario of pure contribution of biogenic emissions, the VOCs were widely dispersed over the Qinling Mountains (Fig. 5d), with a calculated 9.9 ppb for the GZ basin and 12.4 ppb for urban Xi'an. $NO_x$ concentration has the similar pattern as VOCs in the scenario of pure contribution of anthropogenic emissions, with averaged concentrations of 11.0 ppb for the GZ basin and 30.3 ppb for urban Xi'an

(Fig. 6c). However, in the GZ basin and urban Xi'an, biogenic sources contributed less than 0.2 ppb to $NO_x$ concentration (Fig. 6d). In the scenario of pure contribution of anthropogenic emissions, was $PM_{2.5}$ spread over a wider area (Fig. 8c), due to the time required for secondary aerosol formation. In the scenario of pure contribution of biogenic emissions, $PM_{2.5}$ was mostly distributed among the Qinling Mountains (Fig.

8d), but the concentration was lower by one order of magnitude (Table 5, Fig. S4).

In the scenario of pure contribution of anthropogenic emissions, daily peak $O_3$ accumulated in the downwind region in the center and western GZ due to high VOCs and $NO_x$ concentrations. Daily peak $O_3$ concentrations reached 22.6 ppb for GZ and 19.1 ppb for urban Xi'an (Fig. 10c). Lower daily peak $O_3$ concentration was found

outside of GZ basin where less anthropogenic VOCs and $NO_x$ was emitted. In contrast, daily peak $O_3$ was negligible (less than 3 ppb) in the scenario of pure contribution of biogenic emissions (Fig. 10d) due to the low $NO_x$ emissions. However, the



distribution of 24h averaged $O_3$ was different from daily peak $O_3$. 24h averaged $O_3$ concentration in the scenario of pure contribution of anthropogenic emissions was more diluted in the GZ city cluster than for surrounding areas (Fig. 11c). Due to the abundant NO emission and its titration effect on $O_3$, the pure effect of anthropogenic

sources was negative, calculated to be -2.2 ppb in urban Xi'an.

## 4.3 Synergistic impact of the interaction between anthropogenic and biogenic sources

The synergistic impact on $O_3$ formation includes the interactions between
anthropogenic and biogenic sources. In other word, it reflects the potential production trend of either "$O_3$-promoted" or "$O_3$-suppressed" under the natural coexistence of all emission sources. In the cases of $NO_x$, VOCs and $PM_{2.5}$, the synergistic impacts contributed less than ±3% of total concentrations (Fig. 5b, 6b, 8b, S4, and Table 5). However, the synergistic impact on $O_3$ played a remarkable role showing positive
impacts for both daily peak (Fig. 10b) and 24h averaged $O_3$ (Fig. 11b). It means that the mixed state of anthropogenic and biogenic sources potentially enhanced the $O_3$ production more than each single source. To make it more specific, we assumed the original state was the result from ANTH simulation without the biogenic sources (Table 3). Figure 12b shows the original state $O_3$ production regime in the simulated
region. VOC-controlled $O_3$ production regime covered the west of urban Xi'an and the southeast of the GZ basin. In the rest of the GZ basin and the neighboring Shanxi provinces, the $O_3$ production was in the transition regime, controlled by both $NO_x$ and VOC. $NO_x$-controlled $O_3$ production regime dominated the rest of the region. After we included biogenic VOC emissions in the simulation, the $O_3$ concentration was
significantly enhanced in the VOC-controlled regions, and partly enhanced in the mix-controlled region. However, in the VOC-controlled region, the synergistic impact contributed little.

The synergistic impact is of great importance, approximately the same magnitude as the impact from pure contributions of anthropogenic sources. The synergistic
impact contributed daily peak $O_3$ concentrations of 10.5 ppb for the GZ basin and 14.3 ppb for urban Xi'an, while the pure anthropogenic impact contributed 22.6 ppb for GZ basin and 19.1 ppb for urban Xi'an. However, the extent was ~50% smaller on


the 24h averaged scale, but still increased $O_3$ concentration by 5.8 ppb for the GZ basin and 6.8 ppb for urban Xi'an. Figure 9 shows the diurnal variation of the observed and simulated $O_3$ concentration at Xi'an Jiaotong University, as well as the tested contributing components. Transport dominated $O_3$, constantly contributing

30-40 ppb as background. The impact of pure anthropogenic sources was positive on $O_3$ production during 13:00-19:00 but negative during the rest of the time, and the impact of pure biogenic sources was negligible. Synergistic impact of both anthropogenic and biogenic sources resulted in a positive contribution during 10:00-21:00, comparable to the impact of pure anthropogenic sources.

We evaluated the actual contributions from anthropogenic or biogenic sources. The actual contribution is defined as the combination with both pure and synergistic contributions. Actual contribution from anthropogenic sources to $NO_x$ in urban Xi'an was 29.6 ppb, accounting 98% of total $NO_x$. Similarly, actual contribution from anthropogenic sources to $PM_{2.5}$ was 91.1 μg m$^{-3}$, comprising 89% of total $PM_{2.5}$. In

term of VOCs, the actual impact of the anthropogenic source contributed 26.2 ppb (58%), while the biogenic source contributed 12.2 ppb (27%). Due to the synergistic impact on peak $O_3$, both the actual effect of anthropogenic (33.4 ppb) and biogenic (16.8 ppb) sources were noticeable.

**5.  Conclusions and discussion**

The GZ basin is a representative region in the northwest of China, suffering serious air pollution in recent years. Geographically, the GZ basin borders the northern foot of the Qinling Mountains. For this reason, in addition to the anthropogenic emissions from metropolitan areas, biogenic emissions are of great

importance in the region, especially in warm season with active photochemistry. In this study, we used the WRF-Chem model to simulate $O_3$ in the GZ basin and compared the results to near-surface measurements, with the aim of quantifying the pure and synergistic impacts of anthropogenic and/or biogenic sources on summertime $O_3$ formation. The simulation was driven by the best currently available

inventory of anthropogenic emissions and online calculated biogenic emissions. Near-surface measurements were captured from 6 surface sites among the Qinling





Mountains for biogenic VOCs and one 100-m-high site in the Xi'an city for air quality ($NO_x$, VOCs, $O_3$ and $PM_{2.5}$).

Our model successfully reproduced the observed air quality and meteorological parameters. The biogenic VOCs simulation showed a reasonable agreement. Our
model also well-reproduced the magnitudes and variations of $O_3$, $NO_x$ and $PM_{2.5}$ concentrations excluding rainy days, with normalized mean bias less than ±21%.

We further conducted three scenario simulations to explore the pure and synergistic impacts of anthropogenic and/or biogenic sources on $O_3$ and the precursors, by using the factor separation approach (FSA). The results concluded that, for the
precursors, pure impact of anthropogenic source contributed 99% of $NO_x$, 80% of $PM_{2.5}$, and 33% of VOCs in the GZ basin, and pure impact of biogenic source contributed 40% of VOCs but only 1-5% of $PM_{2.5}$ and $NO_x$. Meanwhile, synergistic impacts from the combination of anthropogenic and biogenic sources did not bring significant changes on $NO_x$, VOCs and $PM_{2.5}$ (less than ±4%). In the case of daily
peak $O_3$, the pure impact of anthropogenic source remained the dominant contributor (19.1 ppb for urban Xi'an), even after anthropogenic particles reduced the $NO_2$ photolysis by up to 60%. The abundant biogenic VOCs from the nearby forests promoted the $O_3$ formation by interaction with anthropogenic $NO_x$, contributing 14.4 ppb to $O_3$ in urban Xi'an. This synergistic impact presented a positive contribution to
$O_3$ production throughout the day and the positive effect was much more prominent during 12:00-19:00.

$O_3$ pollution in China has been raising increasing concerns in recent years. Some scientists hold the view that excessive concentration of $PM_{2.5}$ suppressed the formation of $O_3$ in the past, hiding the problem temporally. However, with the
effective control of $PM_{2.5}$, $O_3$ pollution is manifested. The phenomenon can also be demonstrated by the government control action during G20 summit (The Group of Twenty Finance Ministers and Central Bank Governors) in Hangzhou in 2016. The concentration of $PM_{2.5}$ was depressed sharply under the strict emission control, but $O_3$ concentration was even higher than usual. Better understanding of $O_3$ pollution
sources/sinks and formation mechanisms in high $PM_{2.5}$ exposed area in China will benefit and guide the implementation of $PM_{2.5}/O_3$ cooperative control. Our results suggest that, in big cities geographically close to forest, $O_3$ pollution can be enhanced



by the synergistic impact from the combination of biogenic and anthropogenic sources. The synergistic contribution of each single source to $O_3$ formation cannot be neglected when making pollution control strategies.

**Acknowledgments**

This work was supported by the National Natural Science Foundation of China (41705128), and Opening Project of Shanghai Key Laboratory of Atmospheric Particle Pollution and Prevention (LAP[3]) (FDLAP17003).

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


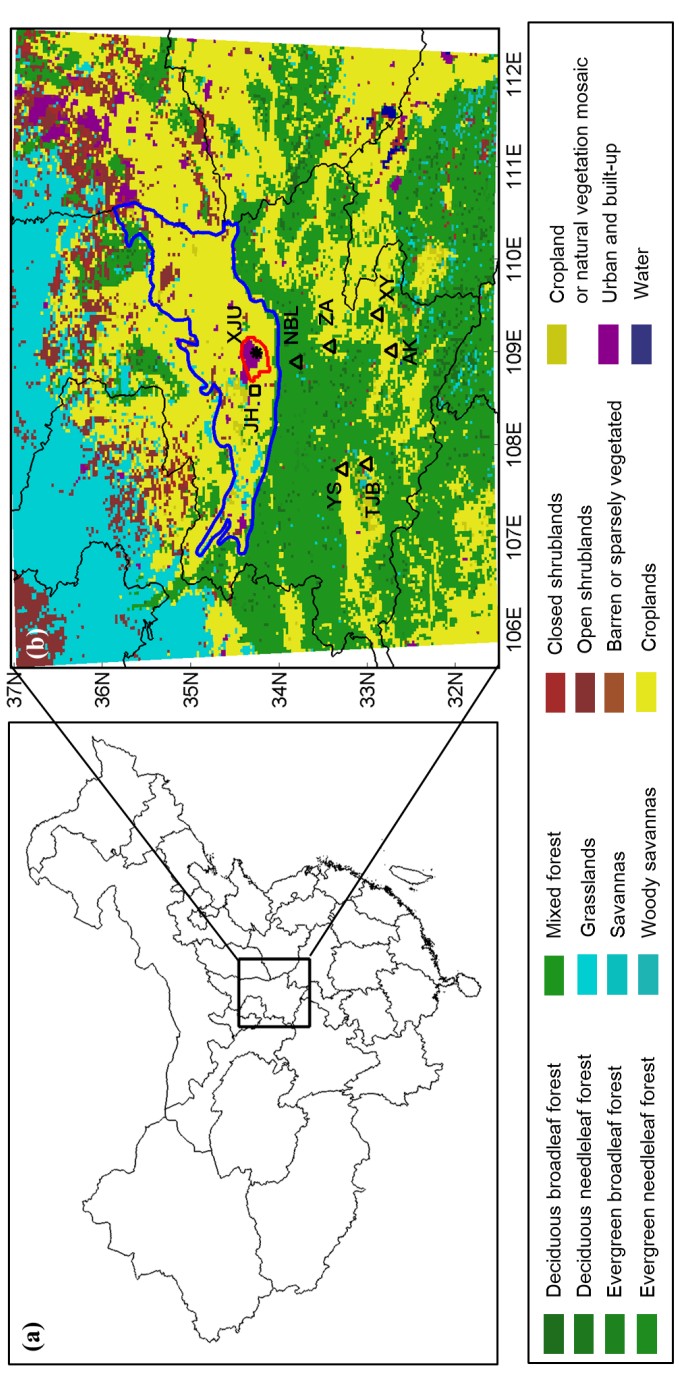

**Figure 1.** The simulation domain and the locations of six biogenic VOCs sites (triangles), one air quality site (snowflake), one meteorological site (square). Underlain are land types from MODIS. The area of red line indicates the urban area of the Xi'an city. The area of blue line indicates the GZ basin.



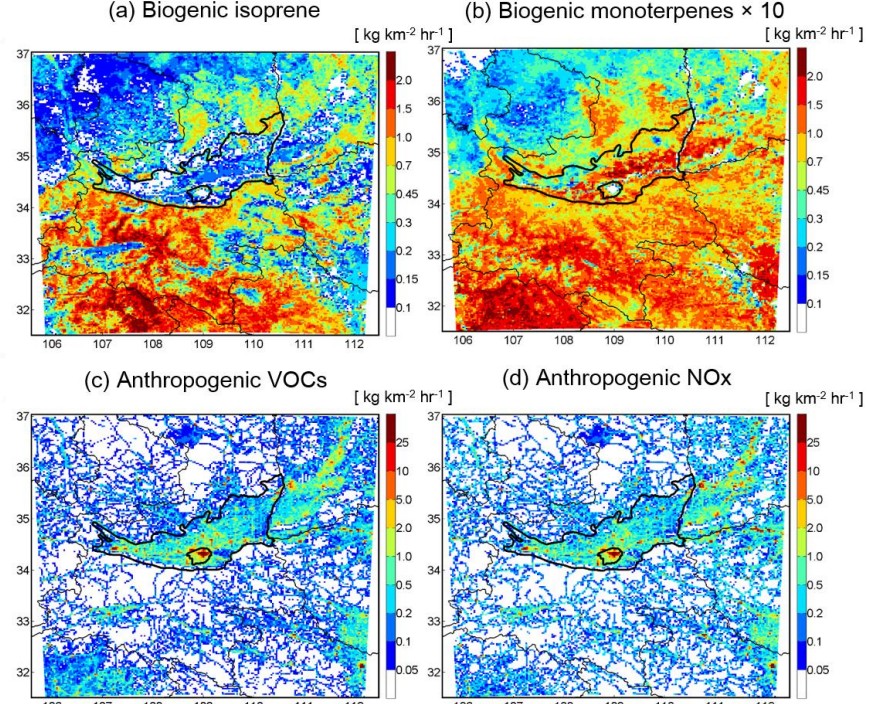

**Figure 2.** Monthly mean emissions of (a) biogenic isoprene, (b) biogenic monoterpenes, (c) anthropogenic VOCs and (d) anthropogenic $NO_x$ in the GZ basin and surrounding areas in August 2011.





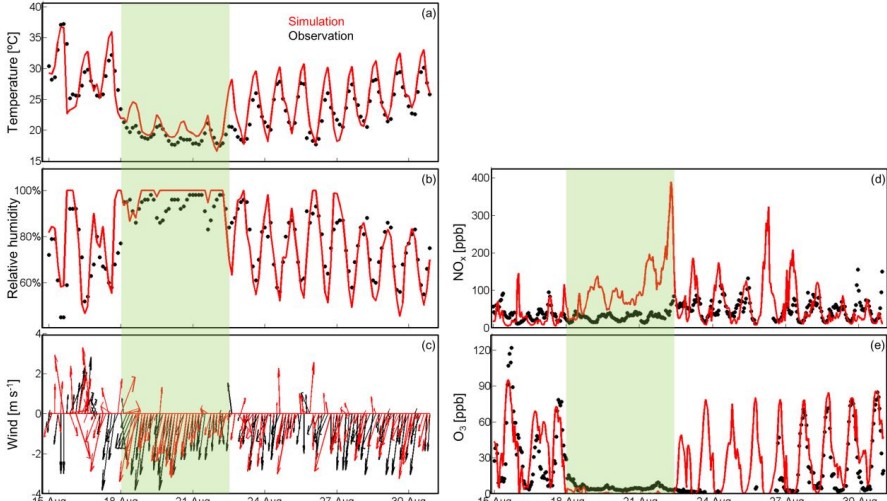

**Figure 3.** Observed (black) and simulated (red) temporal patterns of temperature (a), relative humidity (b) and wind (c) at the Jinghe site and $NO_x$ (d) and $O_3$ (e) concentrations at Xi'an Jiaotong University during the period from 15$^{th}$ to 30$^{th}$ August 2011. The green shadow (18$^{th}$ -22$^{nd}$ August) indicates rainy days.





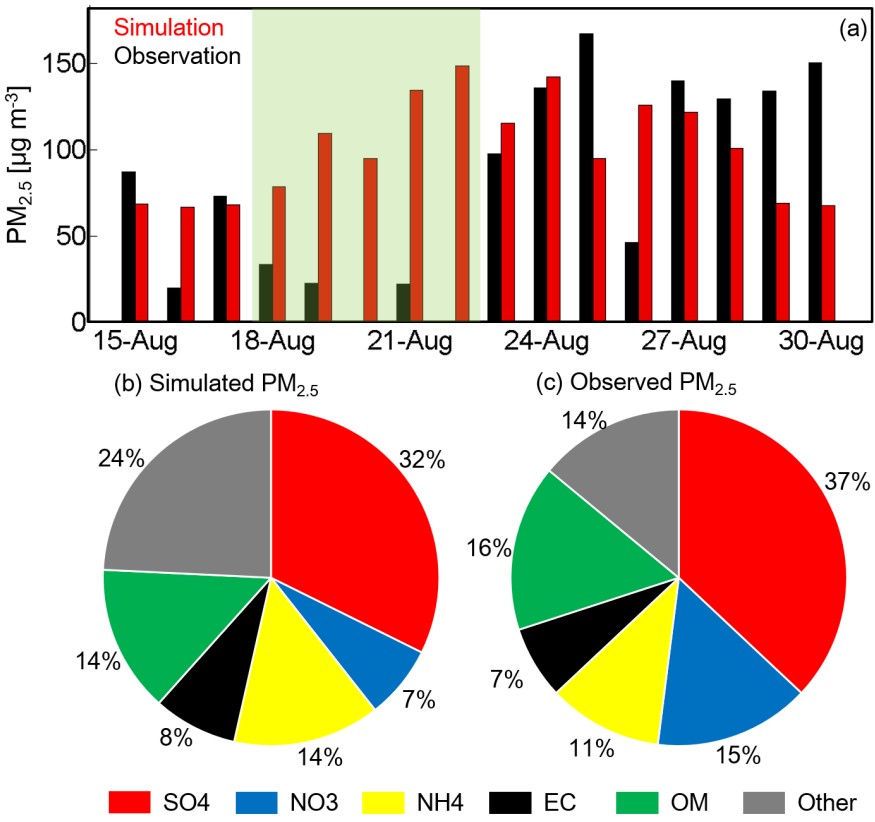

**Figure 4.** Observed and simulated $PM_{2.5}$ concentrations (a) and compositions (b and c) at Xi'an Jiaotong University in August 2011. The components are calculated during the periods excluding the rainy days (the green shadow).





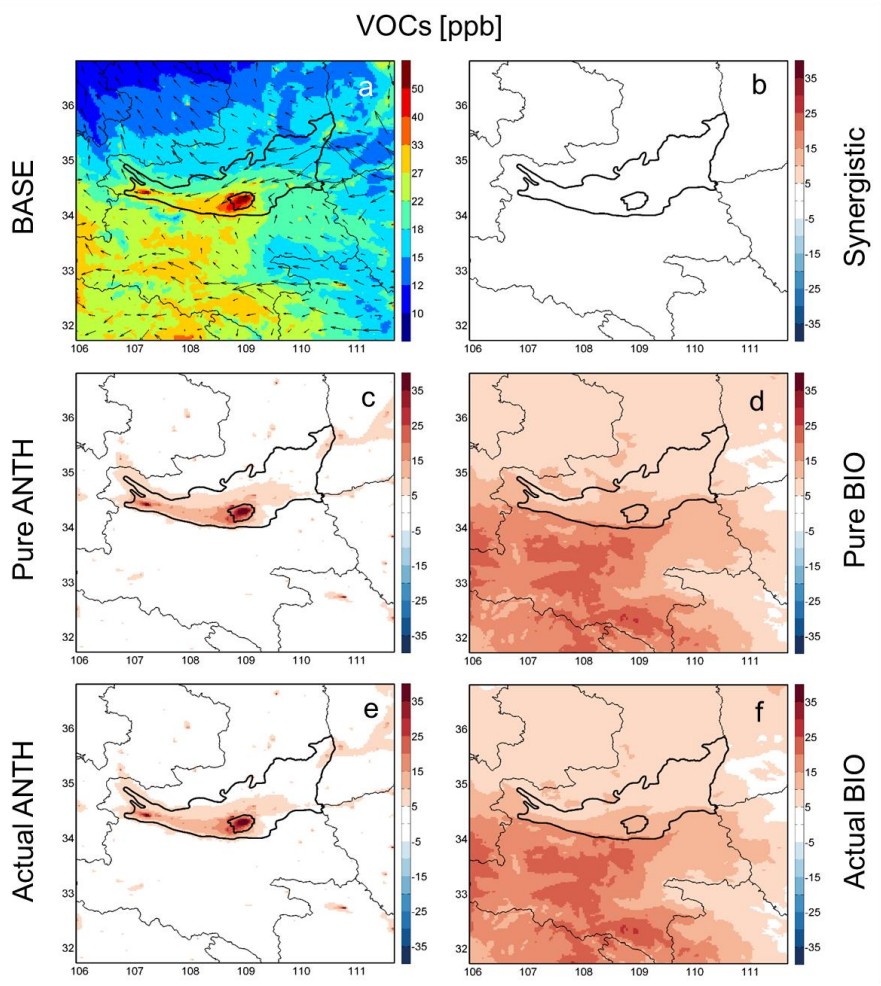

**Figure 5.** Spatial distributions of monthly mean concentrations of VOCs in August 2011. (a) is the result from the BASE simulation, overlaid with simulated wind vectors. (b)-(f) are simulated VOCs concentrations contributed from synergistic anthropogenic and biogenic, pure anthropogenic, pure biogenic, actual anthropogenic and actual biogenic sources, respectively.





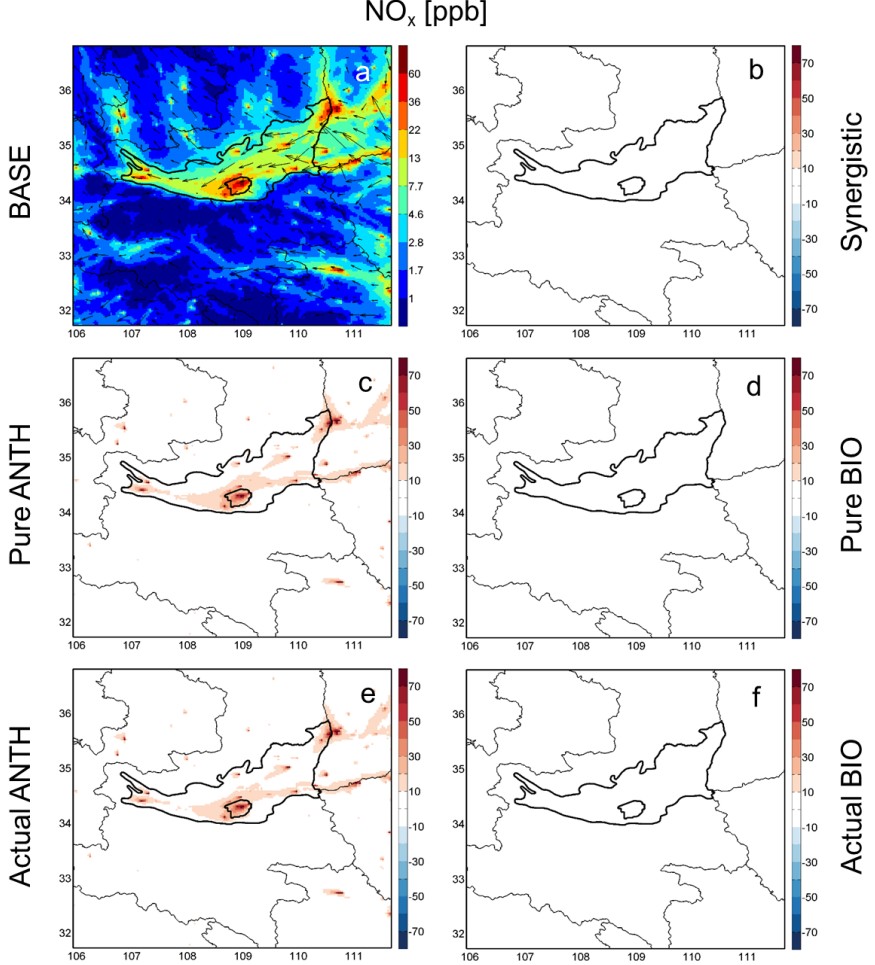

**Figure 6.** Spatial distributions of monthly mean concentrations of $NO_x$ in August 2011. (a) is the result from the BASE simulation, overlaid with simulated wind vectors. (b)-(f) are simulated $NO_x$ concentrations contributed from synergistic anthropogenic and biogenic, pure anthropogenic, pure biogenic, actual anthropogenic and actual biogenic sources, respectively.





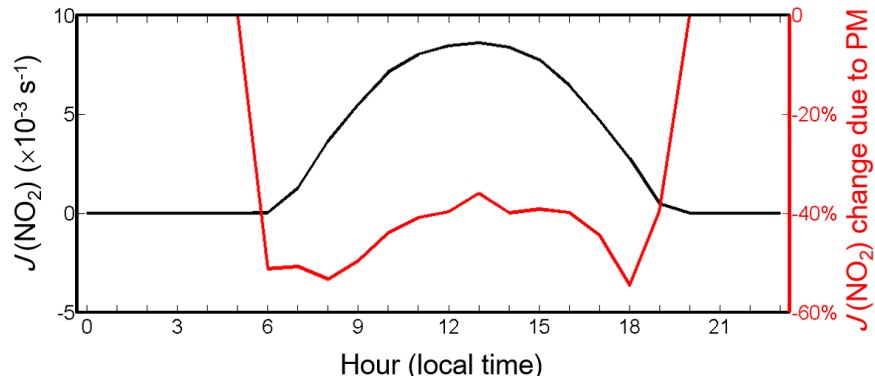

**Figure 7.** Diurnal variations of $J(NO_2)$ (black) and the changes in $J(NO_2)$ (red) averaged in urban Xi'an due to PM effects in August 2011.



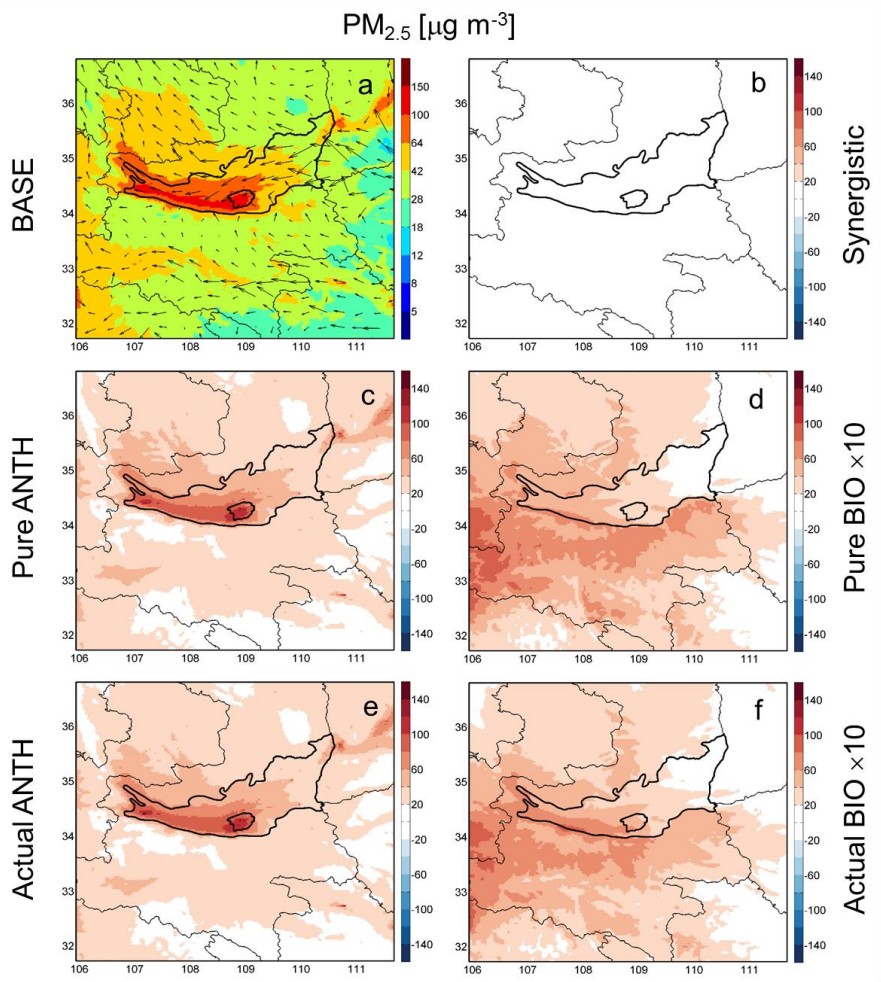

**Figure 8.** Spatial distributions of monthly mean concentrations of PM$_{2.5}$ in August 2011. (a) is the result from the BASE simulation, overlaid with simulated wind vectors. (b)-(f) are simulated PM$_{2.5}$ concentrations contributed from synergistic anthropogenic and biogenic, pure anthropogenic, pure biogenic, actual anthropogenic and actual biogenic sources, respectively.





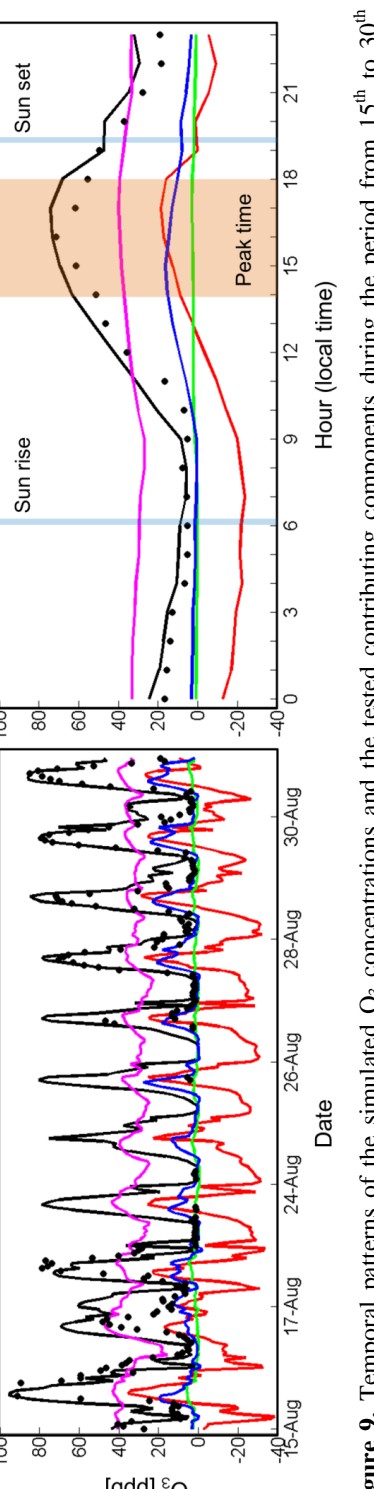

**Figure 9.** Temporal patterns of the simulated $O_3$ concentrations and the tested contributing components during the period from $15^{th}$ to $30^{th}$ August 2011, excluding the rainy days ($18^{th}$ -$22^{nd}$ August). The orange shadow (14:00-18:00) indicates daily $O_3$ peak time.



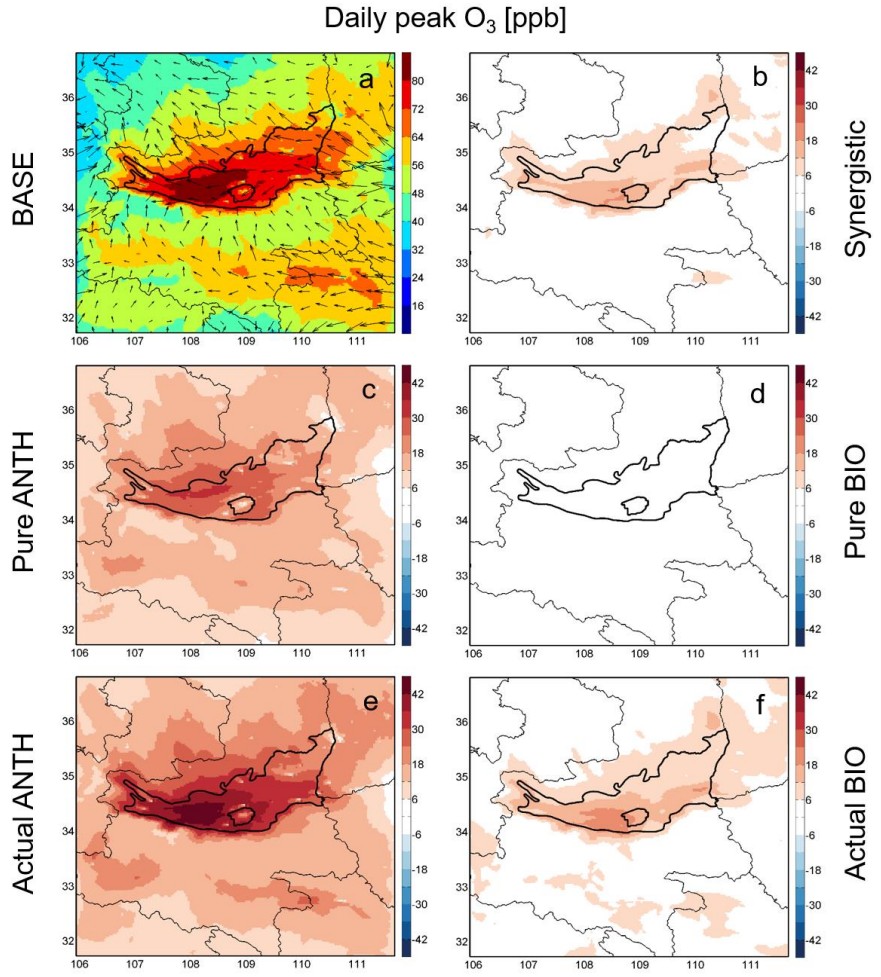

**Figure 10.** Spatial distributions of monthly mean concentrations of daily peak $O_3$ in August 2011. (a) is the result from the BASE simulation, overlaid with simulated wind vectors. (b)-(f) are simulated daily peak $O_3$ concentrations contributed from synergistic anthropogenic and biogenic, pure anthropogenic, pure biogenic, actual anthropogenic and actual biogenic sources, respectively.





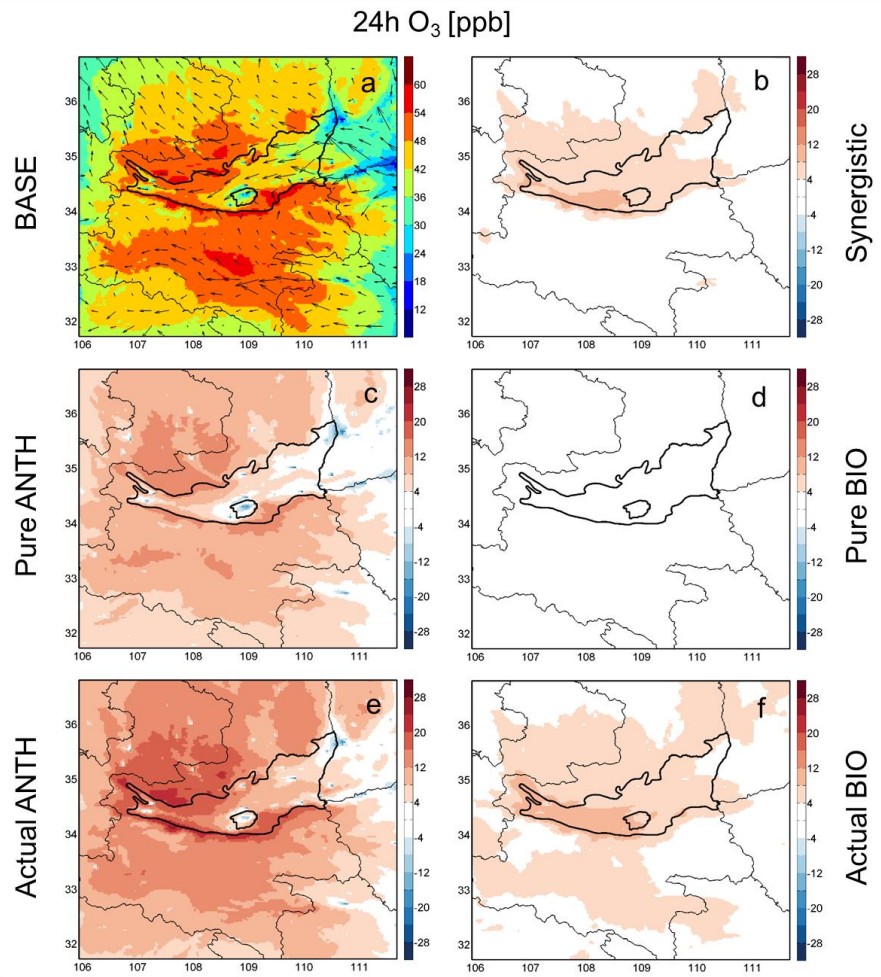

**Figure 11.** Spatial distributions of monthly mean concentrations of 24h averaged $O_3$ in August 2011. (a) is the result from the BASE simulation, overlaid with simulated wind vectors. (b)-(f) are simulated 24h averaged $O_3$ concentrations contributed from synergistic anthropogenic and biogenic, pure anthropogenic, pure biogenic, actual anthropogenic and actual biogenic sources, respectively.





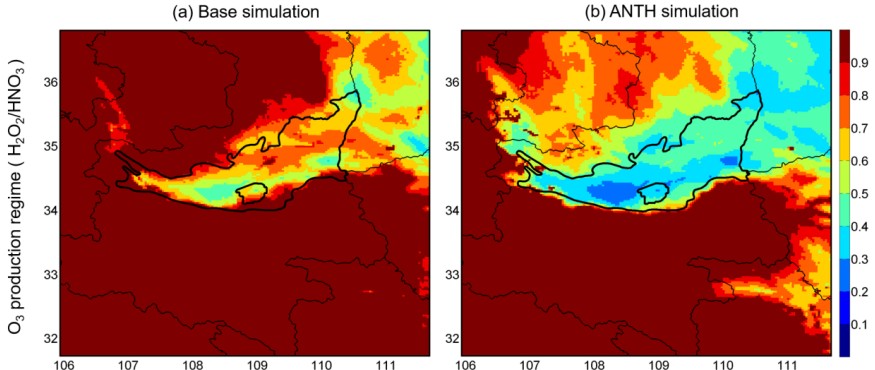

**Figure 12.** The monthly mean ratio of $H_2O_2/HNO_3$ during the daily $O_3$ peak time (14:00-18:00 local time) in August 2011 in the (a) Base simulation and (b) the simulation without biogenic sources.



**Table 1.** Ambient biogenic VOCs observations in the Qinling Mountains during $6^{th} - 7^{th}$ August 2011

| Site [a] | Date | Start Time [b] | Location | Isoprene (ppb) | | Monoterpenes (ppb) | | Dominant monoterpenes [c] |
|---|---|---|---|---|---|---|---|---|
| | | | | Observation | Simulation | Observation | Simulation | |
| NBL | 2011/8/6 | 10:20 | 33.78 °E, 108.88 °N | 3.8 | 4.5 | 0.42 | 0.29 | α-pinene |
| ZA | 2011/8/6 | 12:44 | 33.40 °E, 109.05 °N | 0.1 | 0.9 | 0.16 | 0.22 | α-pinene, limonene, menthone |
| XY | 2011/8/6 | 16:14 | 32.87 °E, 109.40 °N | 1.0 | 0.3 | 0.24 | 0.13 | α-pinene |
| AK | 2011/8/7 | 09:45 | 32.71 °E, 109.01 °N | 0.8 | 0.4 | 0.10 | 0.08 | α-pinene |
| TJB | 2011/8/7 | 13:00 | 32.99 °E, 107.78 °N | 0.5 | 1.1 | 0.27 | 0.26 | α-pinene |
| YS | 2011/8/7 | 14:50 | 33.27 °E, 107.73 °N | 1.6 | 0.9 | 0.04 | 0.32 | α-pinene |
| Average | | | | 1.3 | 1.4 | 0.21 | 0.22 | |

[a] Site names: NBL (Niubeiliang), ZA (Zhenan), XY (Xunyang), AK (Ankang), TJB (Tangjiaba), YS (Youshui)

[b] The sampling duration is 30 minutes.

[c] Other monoterpenes detected (0.1% to 9%) include tricyclene, α-thujene, camphene, sabinene, myrcene, α-phellandrene, Δ-carene, o-cymene, β-ocimene, cineole, isopulegol, isomenthone.



**Table 2.** Domain-wide amount of emissions in August 2011

|  | Anthropogenic (Gg mon$^{-1}$) | Biogenic (Gg mon$^{-1}$) | Total (Gg mon$^{-1}$) |
|---|---|---|---|
| SO$_2$ | 358 | - | 358 |
| NO$_x$ | 110 | 2.1 | 112 |
| NH$_3$ | 69.0 | - | 69.0 |
| PM$_{2.5}$ | 163 | - | 163 |
| VOCs | 72.2 | 204 | 276 |
|     Isoprene | <0.1 | 157 | 157 |
|     Monoterpenes | - | 22.8 | 22.8 |
|     Alkanes | 34.4 | 5.4 | 39.8 |
|     Alkenes | 21.3 | 4.9 | 26.2 |
|     Aromatic | 11.5 | - | 11.5 |
|     Carbonyls | 4.5 | 12.1 | 16.6 |
|     Organic acids | 0.5 | 1.5 | 2.0 |



**Table 3.** Summary of different simulation settings and definitions of the various contributions from anthropogenic and/or biogenic sources.

| Simulation | Simulation results | Anthropogenic emission | Biogenic emission |
|---|---|---|---|
| BASE | $f_{anth\text{-}bio}$ | ✓ | ✓ |
| ANTH | $f_{anth}$ | ✓ | ✗ |
| BIO | $f_{bio}$ | ✗ | ✓ |
| NEITHER | $f_0$ | ✗ | ✗ |
| Contribution | | | |
| $f_{anth\text{-}bio} - f_{bio}$ | Actual contribution of anthropogenic emissions | | |
| $f_{anth\text{-}bio} - f_{anth}$ | Actual contribution of biogenic emissions | | |
| $f'_0 = f_0$ | The contribution of background transport | | |
| $f'_{anth} = f_{anth} - f_0$ | Pure contribution of anthropogenic emissions | | |
| $f'_{bio} = f_{bio} - f_0$ | Pure contribution of biogenic emissions | | |
| $f'_{anth\text{-}bio} = f_{anth\text{-}bio} - (f_{anth} + f_{bio}) + f_0$ | Synergistic contribution of anthropogenic and biogenic emissions | | |





**Table 4.** Statistics of meteorological and air quality variables over the GZ basin in August 2011[a]

| | Mean | | $r$ [d] | NMB [d] | RMSE [d] |
|---|---|---|---|---|---|
| | Observation | Simulation | | | |
| **Meteorology** [b] | | | | | |
| Wind speed (m s$^{-1}$) | 2.6 | 2.5 | - | -6% | 1.8 |
| Temperature (°C) | 25.1 | 24.2 | 0.86 | 4% | 2.5 |
| Relative humidity | 73.6% | 74.2% | 0.72 | 1% | 12% |
| **Air quality** [c] | | | | | |
| NO$_x$ (ppb) | 47.0 | 46.6 | 0.36 | -1% | 18.1 |
| O$_3$ (ppb) | 31.5 | 38.7 | 0.72 | 21% | 8.1 |
| PM$_{2.5}$ (μg m$^{-3}$) | 107 | 94.6 | - | -12% | 49.3 |

[a] Averaged for the period from 15[th] to 30[th] August 2011, excluding the rainy days.

[b] Meteorological data were obtained from the hourly surface measurements at Jinghe station (108.58°E, 34.26°N)

[c] Air quality data were measured at the roof (107 m above ground) of the main building (108.98°E, 34.25°N) on the campus of Xi'an Jiaotong University

[d] r: correlation coefficient; NMB: normalized mean bias; RMSE: root mean square errors.



**Table 5.** The various contribution components of the simulated $O_3$ (and the precursors) and $PM_{2.5}$ in August 2011

| | $NO_x$ | VOCs | $O_3$ [ppb] | | $PM_{2.5}$ |
| | [ppb] | [ppb] | daily peak [a] | 24h | [μg m$^{-3}$] |
|---|---|---|---|---|---|
| ***The GZ basin*** | | | | | |
| Base | 11.1 | 24.5 | 74.1 | 44.4 | 65.1 |
| Pure ANTH | 11.0 | 8.0 | 22.6 | 5.0 | 52.0 |
| Pure BIO | 0.1 | 9.9 | 2.0 | 1.1 | 3.3 |
| Actual ANTH | 10.6 | 7.7 | 33.0 | 10.8 | 53.5 |
| Actual BIO | -0.3 | 9.6 | 12.5 | 7.0 | 4.9 |
| Syn ANTH-BIO | -0.4 | -0.3 | 10.5 | 5.8 | 1.5 |
| ***Urban Xi'an*** | | | | | |
| Base | 30.1 | 44.8 | 74.7 | 38.7 | 102 |
| Pure ANTH | 30.3 | 26.4 | 19.1 | -2.2 | 88.7 |
| Pure BIO | 0.15 | 12.4 | 2.6 | 1.4 | 3.4 |
| Actual ANTH | 29.6 | 26.2 | 33.4 | 4.6 | 91.1 |
| Actual BIO | -0.6 | 12.2 | 16.8 | 8.2 | 5.8 |
| Syn ANTH-BIO | -0.7 | -0.2 | 14.3 | 6.8 | 2.4 |

[a] Daily $O_3$ peak time is from 14:00 to 18:00 local time

