# Peer review of "Nan Li 1,2, Qingyang He 2,3, Jim Greenberg 4, Alex Guenther 5, Jingyi Li 1, Junji Cao 2,6, Jun Wang 7, Hong Liao 1, Qiyuan Wang 2, Qiang Zhang 8"

_Atmospheric Chemistry and Physics, 2017_

## Referee Comment (RC1) · Anonymous Referee #1 · 7 Feb 2018

Review of "Impacts of Biogenic Emissions on Summertime Ozone Formation in the Guanzhong Basin, China," Li et al., ACP (2017)

**Summary**

This study utilizes a regional model to separate the impacts of anthropogenic and biogenic emissions on ozone concentrations in a populated region of central China. It is found that that the synergistic effect of bio + anthro emission is comparable to the anthro. impact along when considering daytime O3 concentrations. The paper is organize fine and the English is mostly OK. The number of figures/tables seems a little high and dilutes the main message (Fig. 9) somewhat with a lot of details. Publication is recommended after consideration of the following minor comments.

**General Comments**

Treatment of uncertainty: Aside from several brief mentions of NMB or similar metrics, there is little discussion of model uncertainties. For example, uncertainties in emission inventories surely propagate into derived O3 partitioning. In particular, it would be helpful to estimate the confidence in the values shown in Fig. 9, which is the key result of the paper.

NOx measurements: the "NOx" measurement uses a hot molybdenum converter, which is known to convert a lot more than NO2 (probably most of NOy). This point may or may not impact the model-measurement comparison shown in Fig. 3 depending on the NOy partitioning, but it should be acknowledged and addressed.

**Specific Comments**

Sect. 2.2.1: It should be made clear hear that only 6 samples total were collected.

Page 7, Line 5: This mechanism seems outdated given the recent leaps in understanding of isoprene chemistry. Of particular concern is treatment of alkyl nitrates (which can be temporary or permanent NOx sinks, depending on the mechanism) and the general assumed fate of Isoprene RO2 radicals. The authors should consider either 1) justifying why this does not impact their results, or 2) assessing potential uncertainties arising from use of an outdated mechanism.

Page 9, Line 5: "Brute force" does not, to the reviewer's knowledge, refer a specific method of source characterization. Please refine or define.

Page 10, Line 22: Does the model use assimilated meteorology? If so and the model is being nudged with observations, this agreement may not be especially remarkable.

Sect. 3.2: There are only 6 observations here, and looking at Table 1 the model seems to have little skill in capturing the variability of those. It would be worthwhile to point this out and justify why that's not a big deal for the present analysis.

Page 11, Line 30: "agree well" seems optimistic. Looking at Fig. 4, 5 of the 11 non-rain days show model-measurement disagreement by more than a factor of 2, and it is not evident that the model captures observed day-to-day variability.

Page 13, Line 7: Is this JNO2 calculation based on model output? Please clarify.

Page 13, Line 15: presumably, aerosol lifetime is also longer than NOx.

Page 15, Line 18: Calling this the "original state" is confusing. If it is the anthropogenic simulation, just refer to it as such.

Page 16, Line 10: It is not clear that the discussion of the "actual" contributions adds much to the message of the paper. Indeed, it is a bit confusing because "pure" and "actual" have similar connotations, and because it convolutes several of the separated contributions.

Sect. 4.3: The lack of a "synergistic" contribution to PM is noteworthy and may deserve a few more sentences of discussion, especially given its importance for air quality. In other regions (e.g. the SE US), there seem to be relatively strong links between anthro/bio emissions and PM.

Table 2: Not sure this is necessary for the story. Could be moved to supplement. Your call.

**Technical Comments**

Page 12, Line 14: discuss

Page 16, Line 20: delete "and discussion"

Fig. 12: suggest modifying colorbar to better separate NOx vs VOC controlled regions (e.g. blue-white-red gradient) and mentioning the limits for each regime in the caption.

---

## Referee Comment (RC2) · Anonymous Referee #2 · 20 Feb 2018

The manuscript presents a modeling effort in assessing the contributions of anthropogenic emissions on ozone (and its precursor) and PM2.5 concentrations in Xi'an, China. The work is well designed, and model results are well presented and analyzed. I would recommend accepting the manuscript with minor revision addressing following concerns.

My major concern with the work comes from the model uncertainty analysis, particularly in the simulation of VOCs and in emission inputs. In my opinion, six off-line VOC measurement points in space and time are not sufficient to validate the model accuracy for the following ozone source apportionment analysis. In addition, an agreement in Aug. 6 & 7 would not necessary indicate an agreement in the rest of month August. The current state of Section 3.2 is insufficient. More material is needed to

show and quantify the model uncertainty in VOC estimation for the area under study. The uncertainty in emission input is needed as it is one of the main sources of model uncertainties.

Another minor comment is with the application of brute-force comparison method in assessing the contributions of different emission sources to ozone concentration. The brute-force method has inherent disadvantages when applied to secondary species such as ozone and PM due to the non-linearity in responses. A critical question I would expect the authors to discuss in the manuscript is about the difference between actual and pure contribution from anthropogenic and biogenic.

Specific questions:

1) In Fig. 7, is the black line based on theoretical calculation or model simulation? It seems to me that the smooth black line is based on the theoretical calculation for clear sky condition. If it was true, then I doubt if the decrease (red line) includes both the cloud and aerosol effects.

2) In Section 3.2, is the simulated isoprene/monoterpene mean concentration the mean over the one-month simulation periods or Aug. 6-7th or the mean of six data points corresponding to the time and location of the six measurements?

3) Why modeled wind speed is discontinuous in Fig. 3c?

---

## Author Comment (AC1) · 10 Mar 2018

**Responses to reviewers:**

We thank the two anonymous reviewers for their helpful comments and suggestions. We have revised the manuscript following the comments as described below. Reviewer comments are shown in blue. Our responses are shown in black. The revised texts are shown in italics.

**Reviewer #1**

(1) Treatment of uncertainty: aside from several brief mentions of NMB or similar metrics, there is little discussion of model uncertainties. For example, uncertainties in emission inventories surely propagate into derived $O_3$ partitioning. In particular, it would be helpful to estimate the confidence in the values shown in Fig. 9, which is the key result of the paper.

Thanks for the comment. We added discussions on uncertainties of (1) emission inventory, (2) chemical mechanism, and (3) biogenic VOC measurements

(1) Emission inventory

Anthropogenic emissions were obtained from the Multi-resolution Emission Inventory for China (Li et al., 2017). Emission estimates from bottom-up inventories are uncertain due to lack of complete knowledge of human activities and emission from different sources. The overall uncertainties of the bottom-up $SO_2$, $NO_x$, $NH_3$, VOC, and $PM_{2.5}$ emission estimates are ±31%, ±37%, ±153%, ±78% and ±133%, respectively, with 95% confidence intervals (CI) (Zhang et al., 2009; Lei et al., 2011; Zhao et al., 2011; Lu et al., 2011; Kurokawa et al., 2013). We updated Table S1 and added text in the Section 2.3 of the revised manuscript to describe this issue.

*The emission estimates and uncertainties of VOCs, $SO_2$, $NO_x$, $NH_3$, and $PM_{2.5}$ in the domain during the simulation period are summarized in Table S1.*

(2) Chemical mechanism

The uncertainty of chemical mechanism is stated in the comment (4).

(3) Biogenic VOCs

The uncertainties of biogenic VOC observation and simulation are stated in the comment (7). In addition, we added discussion about sub-month variability of the simulated biogenic VOCs in Section 3.2 in the revised manuscript.

*We also analyzed the temporal variation of simulated biogenic VOC during the whole simulation period and found the sub-month variability was relatively small (the standard deviation < 25%).*

(2) $NO_x$ measurements: the "$NO_x$" measurement uses a hot molybdenum converter, which is known to convert a lot more than $NO_2$ (probably most of $NO_y$). This point may or may not impact the model measurement comparison shown in Fig.

 depending on the NO$_y$ partitioning, but it should be acknowledged and addressed.

Thank you for pointing this out. The observed NO$_x$ (NO+NO$_2$) concentration in the center of Xi'an was measured by a gas-phase chemiluminescence detection analyzer. The principle of this method is converting ambient NO$_2$ to NO on the hot surface of a molybdenum oxide (MoO) catalyst followed by a chemiluminescence detection of NO. The conversion of the thermal catalyst is not specific to NO$_2$, but also other nitrogen-containing components (NO$_z$: NO$_z$=NO$_y$-NO$_x$, including PANs, HNO$_3$, HO$_2$NO$_2$, HONO, RO$_2$NO$_2$, and organic nitrates), resulting in a higher NO$_2$ concentration. However, despite its drawbacks, it remains a widely used method in air-quality monitoring networks and research, owing in part to its low cost, sensitive detection, and ease of operation.

Xu et al. (2013) evaluated this potential uncertainty of the MoO converter at four different sites in China, and found that the MoO converter worked well at the urban site (overestimation less than 10%), but overestimated NO$_2$ more than 30% at suburban and background sites. In the Guanzhong basin, we used the simulated NO$_y$ to evaluate the uncertainty of the NO$_x$ measurements, due to the lack of referenced NO$_y$ or NO$_x$ observation. The calculated results indicated that the NO$_z$ accounted for 11% of the total NO$_y$ at the urban Xi'an during the no-raining period of Fig. 3. We noted the uncertainty in the NO$_x$ measurements, but thought this would not crucially impact the model-measurement comparison.

We modified the text in Section 2.1.2 and 3.3 in the revised manuscript to describe this issue.

*Section 2.1.2*

*NO$_x$ was measured by a gas-phase chemiluminescence detection analyzer EC9841, coupled with a hot molybdenum converter.*

*Section 3.3*

*It is worth noting that the observed NO$_x$ were detected by a chemiluminescence analyzer coupled with a hot molybdenum converter (MoO), and this method was recognized to cause higher NO$_2$ concentration due to the positive interference of other nitrogen-containing components (NO$_z$, e.g. PAN, HNO$_3$ and HONO). Xu et al. (2013) found that the uncertainty caused by the MoO converter was much smaller at urban sites (less than 10%) than that at suburban and background sites (more than 30%). In the GZ basin, to evaluate the uncertainty, we estimated the ratio of NO$_z$/(NO$_x$+NO$_z$) from the model. The calculated results indicated that the NOz accounted for 11% of the total NO$_x$+NO$_z$ at urban Xi'an during the no-raining period. We noted the uncertainty in our NO$_x$ measurements, but thought this would not crucially impact the model-measurement comparison.*

(3) Sect. 2.2.1: It should be made clear hear that only 6 samples total were collected.

Thanks for the suggestion. We modified the text in Section 2.1.1 in the revised manuscript as follows.

*We selected six field sites in the Qinling Mountains (Fig. 1b, the triangles) and collected one ambient air sample at each site on 6$^{th}$ – 7$^{th}$ August 2011 under sunny*

*weather conditions (details are presented in Table 1).*

(4) Page 7, Line 5: This mechanism seems outdated given the recent leaps in understanding of isoprene chemistry. Of particular concern is treatment of alkyl nitrates (which can be temporary or permanent $NO_x$ sinks, depending on the mechanism) and the general assumed fate of Isoprene $RO_2$ radicals. The authors should consider either 1) justifying why this does not impact their results, or 2) assessing potential uncertainties arising from use of an outdated mechanism.

Thank you for pointing this out. Isoprene is the most important NMVOC (Guenther et al., 2006), affecting tropospheric $O_3$, OH and aerosols in complex ways. Being highly reactive, isoprene has a lifetime of only ~3h at OH concentration of $1 \times 10^6$ mol cm$^{-3}$. In the daytime, isoprene reacts with OH radical to form hydroperoxy radicals (ISOPO2). Subsequently, in the presence of $NO_x$, ISOPO2 reacts with NO leading to the production of hydroxynitrates (ISOPN) by a minor branch, which sequester $NO_x$ and therefore regulate $O_3$ formation locally. With the presence of a double bond, ISOPNs are highly reactive with respect to OH and O3. The products from ISOPN oxidation might either release $NO_x$ or generate secondary organic nitrates. Some of the secondary organic nitrates products are found to be considerably longer-lived than ISOPN and can serve as temporary reservoirs for $NO_x$ (Paulot et al., 2009a, b; 2012). Besides daytime oxidation, nighttime oxidation of isoprene by $NO_3$ contributes significantly to the budget of organic nitrates (Horowitz et al., 2007; von Kuhlmann et al., 2004; Xie et al., 2013).

Many laboratory, filed observations and simulation studies (e.g. von Kuhlmann et al., 2004; Perring et al., 2009; Paulot et al., 2009a) agreed that tropospheric $O_3$ production was highly sensitive to the ISOPN yield (4.4% to 15%) and $NO_x$ recycling efficiency (3% to 50%). However, large uncertainties still remain in estimating these parameters. Horowitz et al. (2007) found a 4% ISOPN yield best captured the alkyl and multifunctional nitrates measured by aircraft, and unreasonably high ISOPN yield (18%) would let ISOPN to be a terminal sink for $NO_x$ (Hudman et al., 2009).

Some recent modeling studies evaluated the sensitivity of tropospheric $O_3$ to isoprene oxidation chemistry. Xie et al. (2013) incorporated recent advances in isoprene oxidation chemistry in CMAQ (including a more explicit representation of isoprene nitrate formation from OH/NO and $NO_3$ pathways as well as modification to the isoprene chemistry under low-NOx conditions) and found the model can capture observations by assuming the ISOPN yield of 6% to 12%. O3 increased by 2 ppbv in the eastern U.S as a result of these updates. Also focusing on the eastern U.S., Mao et al. (2013) implemented an updated isoprene chemistry mechanism in GEOS-Chem and found $O_3$ increased by 3-5 ppbv.

We added the discussion above in Section 2.2 to state the uncertainty.

*We noted some advances in isoprene nitrate chemistry in recent year. A number of laboratory, filed observation and simulation studies (e.g. Paulot et al., 2009a, b, 2012; Horowitz et al., 2007; Hudman et al., 2009) highlighted the importance of the yield of isoprene nitrate (4-15%), the $NO_x$ recycling efficiency (3-50%) and the representation of secondary nitrate photochemistry for simulation of tropical $O_3$. They all agreed there were still large uncertainties in isoprene nitrate chemistry. Some recent modeling studies evaluated the sensitivity of tropospheric $O_3$ to isoprene*

*oxidation chemistry. Xie et al. (2013) incorporated recent advances in isoprene oxidation chemistry in CMAQ and found the model can capture observations by assuming the isoprene nitrate yield of 6% to 12%. Simulated $O_3$ increased by 2 ppbv in the eastern U.S. as a result of these updates. Also focusing on the eastern U.S., Mao et al. (2013) implemented an updated isoprene chemistry mechanism in GEOS-Chem and found $O_3$ increased by 3-5 ppbv.*

(5) Page 9, Line 5: "Brute force" does not, to the reviewer's knowledge, refer a specific method of source characterization. Please refine or define.

Thank you for the comment. We defined the "Brute force" method in Section 2.4 in the revised manuscript as follows.

*The approach referred to as the "brute-force" method (sensitivity analysis used to measure the model output response to emission changes) is traditionally used in air quality model to identify source contributions from specific non-reactive species in a linear process....*

(6) Page 10, Line 22: Does the model use assimilated meteorology? If so and the model is being nudged with observations, this agreement may not be especially remarkable.

We appreciate the reviewer's concern. We did not include assimilated meteorology in this study.

(7) Sect. 3.2: There are only 6 observations here, and looking at Table 1 the model seems to have little skill in capturing the variability of those. It would be worthwhile to point this out and justify why that's not a big deal for the present analysis.

We agree that our model didn't well capture the variability of the observed BVOCs. We think it is mainly because that, in our 3×3 km grid, different terpene emitters are not homogeneously distributed in the grid and the point measurements are influenced by the microenvironment and meteorology (Zare et al., 2012; Kota et al., 2015).

However, in this study, our goal is estimating the biogenic effect on $O_3$ formation in an urban area 50 km away from the foothills. We pay more attention to the magnitude of BVOC concentration at the regional scale, instead of capturing the hourly- and microenvironment-scale variability in either the observation or the simulation. Therefore, we think it is reasonable to compare the averaged simulations of 6 grids with the measurements.

We modified the text in Section 3.2 of the revised manuscript to state the uncertainty and to describe the above rationale.

*In general, different terpene emitters are not homogeneously distributed in a kilometer-scale grid and the point measurements are influenced by the microenvironment and meteorology (Zare et al., 2012; Kota et al., 2015). However, in this study, our goal is to estimate the biogenic effects on urban $O_3$ 50 km away from*

*the foothills, which requests more concern on the regional scale VOC level, rather than the microenvironment-scale variability in either the observation or the simulation. Thus, we compared the average of VOC measurements with model simulations to validate whether the calculated results were reasonable. The isoprene mean concentration simulated in the six grids (corresponding to the time of observations) was 1.4 ppb, which is close to the observed average value of 1.3 ppb at the six sampling sites. Monoterpenes performed quite similarly, simulated 0.22 ppb comparing with observed 0.21 ppb. We also analyzed the temporal variation of simulated biogenic VOC during the whole simulation period and found the sub-month variability was relatively small (the standard deviation < 25%). The evaluation indicates that biogenic VOCs simulations reasonably agreed with the observations in the Qinling Mountains, on average, which provides a basis for us to further evaluate biogenic effects on $O_3$.*

(8) Page 11, Line 30: "agree well" seems optimistic. Looking at Fig. 4, 5 of the 11 non-rain days show model measurement disagreement by more than a factor of 2, and it is not evident that the model captures observed day-to-day variability.

Thank you for pointing this out. We modified text in Section 3.3 in the revised manuscript to make our statements more precise.

*The model predicted $PM_{2.5}$ concentration to be 94.6±28.2 μg m$^{-3}$, slightly lower (NMB=-12%) than measured 107 μg m$^{-3}$ averaged for the no-raining period, but didn't perform well in capturing temporal correlation (r=0.17). The simulated $PM_{2.5}$ showed the similar compositions to the observation (Fig. S1b and S1c)....*

(9) Page 13, Line 7: Is this $JNO_2$ calculation based on model output? Please clarify.

Yes, the $J(NO_2)$ is calculated by model output "photor_no2". We added the text in Section 4.1 in the revised manuscript as follows.

*Figure 6 shows the changes of $J(NO_2)$ (calculated by model track output photor_no2)with the participation of PM averaged for urban Xi'an.*

(10) Page 13, Line 15: presumably, aerosol lifetime is also longer than $NO_x$.

Thank you for the suggestion. We added this in Section 4.1 in the revised manuscript.

*The spatial distribution of high-values of $PM_{2.5}$ was similar to that of $NO_x$, but covered a wider area mostly in the downwind region of urban Xi'an, which is expected due to longer lifetime of aerosols compared with $NO_x$ and the time required for secondary aerosol formation, thus further dispersion.*

(11) Page 15, Line 18: Calling this the "original state" is confusing. If it is the anthropogenic simulation, just refer to it as such.

Thank you for the comment. We modified the text in Section 4.3 in the revised

manuscript to make our statements clearer.

*To make it more specific, we started the discussion from the result of ANTH simulation without the biogenic sources (Table 2). Figure 10b shows the $O_3$ production regime in the ANTH simulation.*

(12) Page 16, Line 10: It is not clear that the discussion of the "actual" contributions adds much to the message of the paper. Indeed, it is a bit confusing because "pure" and "actual" have similar connotations, and because it convolutes several of the separated contributions.

Thank you for the comment. We deleted the discussion of the actual contributions to avoid ambiguity.

(13) Sect. 4.3: The lack of a "synergistic" contribution to PM is noteworthy and may deserve a few more sentences of discussion, especially given its importance for air quality. In other regions (e.g. the SE US), there seem to be relatively strong links between anthro/bio emissions and PM.

Thank you for the suggestion. We added the text in Section 4.3 in the revised manuscript to discuss this issue.

*It is worth noting that the biogenic contribution to $PM_{2.5}$ is not obvious (less than 3%) in GZ basin, which might be different from some other regions (e.g. Fu et al., 2012; Li et al., 2013). The main reasons are that 1) organic matter, the most important biogenic $PM_{2.5}$ component, only accounted for 14-16% of $PM_{2.5}$ in GZ basin in August; 2) Undeniably, uncertainties still exist in organic matter simulations in the model.*

(14) Table 2: Not sure this is necessary for the story. Could be moved to supplement. Your call.

Thanks for the suggestion. We moved this table to the supplement of the revised manuscript.

(15) Page 12, Line 14: discuss

Thanks. Corrected

(16) Page 16, Line 20: delete "and discussion"

Thanks. Corrected

(17) Fig. 12: suggest modifying colorbar to better separate NOx vs VOC controlled regions (e.g. blue-whitered gradient) and mentioning the limits for each regime in the caption.

Thank you for the suggestion. We modified the colorbar of Fig. 10 in the revised manuscript.

[Figure]

**Figure 10.** The monthly mean ratio of $H_2O_2/HNO_3$ during the daily $O_3$ peak time (14:00-18:00 local time) in August 2011 in the (a) Base simulation and (b) the simulation without biogenic sources.

**Reviewer #2**

(1) My major concern with the work comes from the model uncertainty analysis, particularly in the simulation of VOCs and in emission inputs. In my opinion, six off-line VOC measurement points in space and time are not sufficient to validate the model accuracy for the following ozone source apportionment analysis. In addition, an agreement in Aug. 6 & 7 would not necessary indicate an agreement in the rest of month August. The current state of Section 3.2 is insufficient. More material is needed to show and quantify the model uncertainty in VOC estimation for the area under study. The uncertainty in emission input is needed as it is one of the main sources of model uncertainties.

Thanks for the comment. We added discussions about uncertainties of emission inventory and biogenic VOC simulations.

(1) Biogenic VOC

We agree that only six off-line measurement points are not sufficient to describe the spatial and temporal variability of BVOCs. However, in this study, our goal is estimating the biogenic effect on $O_3$ formation in urban Xi'an. We pay more attention to the magnitude of BVOC concentration at the regional scale, instead of capturing the hourly- and microenvironment-scale variability in either the observation or the simulation.

To verify the representativeness of BVOC observation (in Aug. 6 & 7), we analyzed the magnitude and temporal variability of simulated BVOC in the Qinling Mountains for the whole simulation period. The results indicated that the mean BVOC (isoprene + monoterperens) concentration is 1.73 ppb, close to the observed value (1.51 ppb) during Aug 6 & 7, and the sub-month variability (characterized by

the standard deviation of daily mean simulation (averaged for 08:00-16:00 local time)) were <25%, indicating that the sub-month variability was relatively small.

We modified the text in the Section 3.2 of the revised manuscript to state the uncertainty.

*In general, different terpene emitters are not homogeneously distributed in a kilometer-scale grid and the point measurements are influenced by the microenvironment and meteorology (Zare et al., 2012; Kota et al., 2015). However, in this study, our goal is to estimate the biogenic effects on urban $O_3$ 50 km away from the foothills, which requests more concern on the regional scale VOC level, rather than the microenvironment-scale variability in either the observation or the simulation. Thus, we compared the average of VOC measurements with model simulations to validate whether the calculated results were reasonable. The isoprene mean concentration simulated in the six grids (corresponding to the time of observations) was 1.4 ppb, which is close to the observed average value of 1.3 ppb at the six sampling sites. Monoterpenes performed quite similarly, simulated 0.22 ppb comparing with observed 0.21 ppb. We also analyzed the temporal variation of simulated biogenic VOC during the whole simulation period and found the sub-month variability was relatively small (the standard deviation < 25%). The evaluation indicates that biogenic VOCs simulations reasonably agreed with the observations in the Qinling Mountains, on average, which provides a basis for us to further evaluate biogenic effects on $O_3$.*

(2) Emission inventory

Anthropogenic emissions were obtained from the Multi-resolution Emission Inventory for China (Li et al., 2017). Emission estimates from bottom-up inventories are uncertain due to lack of complete knowledge of human activities and emission from different sources. The overall uncertainties of the bottom-up $SO_2$, $NO_x$, $NH_3$, VOC, and $PM_{2.5}$ emission estimates are ±31%, ±37%, ±153%, ±78% and ±133%, respectively, with 95% confidence intervals (CI) (Zhang et al., 2009; Lei et al., 2011; Zhao et al., 2011; Lu et al., 2011; Kurokawa et al., 2013). We updated Table S1 and added text in Section 2.3 of the revised manuscript to describe this issue.

*The emission estimates and uncertainties of VOCs, $SO_2$, $NO_x$, $NH_3$, and $PM_{2.5}$ in the domain during the simulation period are summarized in Table S1.*

(2) Another minor comment is with the application of brute-force comparison method in assessing the contributions of different emission sources to ozone concentration. The brute-force method has inherent disadvantages when applied to secondary species such as ozone and PM due to the non-linearity in responses. A critical question I would expect the authors to discuss in the manuscript is about the difference between actual and pure contribution from anthropogenic and biogenic.

The sensitivity analysis (brute-force method) used to measure the model output response to emission changes, is traditionally used in air quality model to identify source contributions from non-reactive species in a linear process. However, as the reviewer mentioned, it is not proper to secondary species such as $O_3$ and PM, because secondary species are generated nonlinearly, and interactions between different

sources cannot be ignored. The difference between the actual and pure contribution from anthropogenic or biogenic source is the synergistic effect between the two sources. In this work, we combined the brute-force method and Factor Separation Technique (FST) (section 2.4) to discuss the importance of synergistic effect on $O_3$ formation. We modified the text in the Section 2.4 in the revised manuscript to make our statement clearer.

*$O_3$ is formed by complicated nonlinear reactions of anthropogenic and biogenic precursors ($NO_x$ and VOCs) in the presence of sunlight. The approach referred to as the "brute-force" method (sensitivity analysis used to measure the model output response to emission changes) is traditionally used in air quality model to identify source contributions from specific non-reactive species in a linear process, but it cannot straightforward apply to secondary species due to the non-linearity in responses. In practice, the actual impact of one factor in a nonlinear process in the presence of others can be separated into 1) pure impact from the factor and 2) interactional impacts from all those factors. In this study, we adopted the factor separation approach (FSA) (Stein and Alpert, 1993) to decompose the pure contribution of a factor from its interaction with other factors.*

(3) In Fig. 7, is the black line based on theoretical calculation or model simulation? It seems to me that the smooth black line is based on the theoretical calculation for clear sky condition. If it was true, then I doubt if the decrease (red line) includes both the cloud and aerosol effects.

The black line is based on model simulated output "photor_no2". We modified the text in Section 4.1 in the revised manuscript to make this issue clear.

*Figure 6 shows the changes of $J(NO_2)$ (calculated by model track output photor_no2) under the participation of PM averaged for urban Xi'an.*

(4) In Section 3.2, is the simulated isoprene/monoterpene mean concentration the mean over the one-month simulation periods or Aug. 6-7th or the mean of six data points corresponding to the time and location of the six measurements?

The simulated isoprene and monoterpenes concentrations in Section 3.2 were the mean of six data points corresponding to the time and location of the six measurements. We modified the text in Section 3.2 in the revised manuscript to make the statement clearer.

*The isoprene mean concentration simulated in the six grids (corresponding to the time of observations) was 1.4 ppb, which is close to the observed average value of 1.3 ppb at the six sampling sites.*

(5) Why modeled wind speed is discontinuous in Fig. 3c?

We appreciate the reviewer's concern. Figure 3c is presented as a "feather picture", in which arrow direction indicates wind direction and length of arrow indicates wind speed. We give a line graph as follow to show wind speed more clearly.

[Figure]

**Reference**

Fu, T.M., Cao, J.J., Zhang, X.Y., Lee, S.C., Zhang, Q., Han, Y.M., Qu, W.J., Han, Z., Zhang, R., Wang, Y.X., Chen, D., and Henze, D. K.: Carbonaceous aerosols in China: top-down constraints on primary sources and estimation of secondary contribution, Atmos. Chem. Phys., 12, 2725-2746, doi:10.5194/acp-12-2725-2012, 2012.

Guenther, A., Karl, T., Harley, P., Wiedinmyer, C., Palmer, P.I., and Geron, C.: Estimates of global terrestrial isoprene emissions using MEGAN (Model of Emissions of Gases and Aerosols from Nature), Atmos. Chem. Phys., 6, 3181-3210, doi:10.5194/acp-6-3181-2006, 2006.

Horowitz, L. W., Fiore, A.M., Milly, G.P., Cohen, R.C., Perring, A., Wooldridge, P.J., Hess, P.G., Emmons, L.K., and Lamarque J.F.: Observational constraints on the chemistry of isoprene nitrates over the eastern United States, J. Geophys. Res., 112, D12S08, doi:10.1029/2006JD007747, 2007.

Hudman, R.C., Murray, L.T., Jacob, D.J., Turquety, S., Wu, S., Millet, D.B., Avery, M., Goldstein, A.H., and Holloway, J.: North American influence on tropospheric ozone and the effects of recent emission reductions: Constraints from ICARTT observations, J. Geophys. Res., 114, D07302, doi:10.1029/2008JD010126, 2009.

Kota, S. H., Schade, G., Estes, M., Boyer, D., and Ying, Q.: Evaluation of MEGAN predicted biogenic isoprene emissions at urban locations in Southeast Texas, Atmospheric Environment, 110, 54-64, 10.1016/j.atmosenv.2015.03.027, 2015.

Kurokawa, J., Ohara, T., Morikawa, T., Hanayama, S., Janssens-Maenhout, G., Fukui, T., Kawashima, K., and Akimoto, H.: Emissions of air pollutants and greenhouse gases over Asian regions during 2000-2008: Regional Emission inventory in ASia (REAS) version 2, Atmos. Chem. Phys., 13, 11019-11058, doi:10.5194/acp-13-11019-2013, 2013.

Lei, Y., Zhang, Q., He, K. B., and Streets, D. G.: Primary anthropogenic aerosol emission trends for China, 1990-2005, Atmos. Chem. Phys., 11, 931-954,

doi:10.5194/acp-11-931-2011, 2011.

Li, M., Zhang, Q., Kurokawa, J.I., Woo, J.H., He, K., Lu, Z., Ohara, T., Song, Y., Streets, D.G., Carmichael, G.R., Cheng, Y., Hong, C., Huo, H., Jiang, X., Kang, S., Liu, F., Su, H., and Zheng, B.: MIX: a mosaic Asian anthropogenic emission inventory under the international collaboration framework of the MICS-Asia and HTAP, Atmos. Chem. Phys., 17, 935-963, doi:10.5194/acp-17-935-2017, 2017.

Li, N., Fu, T.-M., Cao, J., Lee, S., Huang, X.-F., He, L.-Y., Ho, K.-F., Fu, J. S., and Lam, Y.-F.: Sources of secondary organic aerosols in the Pearl River Delta region in fall: Contributions from the aqueous reactive uptake of dicarbonyls, Atmos. Environ., 76, 200-207, doi:10.1016/j.atmosenv.2012.12.005, 2013.

Lu, Z., Zhang, Q., and Streets, D. G.: Sulfur dioxide and primary carbonaceous aerosol emissions in China and India, 1996-2010, Atmos. Chem. Phys., 11, 9839-9864, doi:10.5194/acp-11-9839-2011, 2011.

Mao, J., Paulot, F., Jacob, D. J., Cohen, R.C., Crounse, J.D., Wennberg, P.O., Keller, C.A., Hudman, R.C., Barkley, M.P., and Horowitz, L.W.: Ozone and organic nitrates over the eastern United States: Sensitivity to isoprene chemistry, J. Geophys. Res., 118, 11256-11268, doi:10.1002/jgrd.50817, 2013.

Paulot, F., Crounse, J.D., Kjaergaard, H.G., Kroll, J.H., Seinfeld, J.H., and Wennberg, P.O.: Isoprene photooxidation: New insights into the production of acids and organic nitrates, Atmos. Chem. Phys., 9(4), 1479–1501, 2009a.

Paulot, F., Crounse, J.D., Kjaergaard, H.G., Kurten, A., St Clair, J.M., Seinfeld, J.H., and Wennberg, P.O.: Unexpected epoxide formation in the gas-phase photooxidation of isoprene, Science, 325(5941), 730–733, 2009b.

Paulot, F., Henze, D.K., and Wennberg, P.O.: Impact of the isoprene photochemical cascade on tropical ozone, Atmos. Chem. Phys., 12(3), 1307–1325, 2012.

Perring, A.E., Wisthaler, A., Graus, M., Wooldridge, P.J., Lockwood, A.L., Mielke, L.H., Shepson, P.B., Hansel, A., and Cohen, R.C.: A product study of the isoprene + NO3 reaction, Atmos. Chem. Phys., 9(14), 4945–4956, 2009.

von Kuhlmann, R., Lawrence, M.G., Poschl, U., and Crutzen, P.J.: Sensitivities in global scale modeling of isoprene, Atmos. Chem. Phys., 4, 1–17, 2004.

Xie, Y., Paulot, F., Carter, W.P.L., Nolte, C.G., Luecken, D.J., Hutzell, W.T., Wennberg, P.O., Cohen, R.C., and Pinder R.W.: Understanding the impact of recent advances in isoprene photooxidation on simulations of regional air quality, Atmos. Chem. Phys., 13(16), 8439–8455, 2013.

Xu, Z., Wang, T., Xue, L. K., Louie, P. K. K., Luk, C. W. Y., Gao, J., Wang, S. L., Chai, F. H., and Wang, W. X.: Evaluating the uncertainties of thermal catalytic conversion in measuring atmospheric nitrogen dioxide at four differently polluted sites in China, Atmos. Environ., 76, 221-226, doi: 10.1016/j.atmosenv.2012.09.043, 2013.

Zhang, Q., Streets, D. G., Carmichael, G. R., He, K. B., Huo, H., Kannari, A., Klimont, Z., Park, I. S., Reddy, S., Fu, J. S., Chen, D., Duan, L., Lei, Y., Wang, L. T., and Yao, Z. L.: Asian emissions in 2006 for the NASA INTEX-B mission, Atmos. Chem. Phys., 9, 5131-5153, doi:10.5194/acp-9-5131-2009, 2009.

Zare, A., Christensen, J. H., Irannejad, P., and Brandt, J.: Evaluation of two isoprene emission models for use in a long-range air pollution model, Atmos Chem Phys, 12, 7399-7412, 10.5194/acp-12-7399-2012, 2012.

Zhao, Y., Nielsen, C. P., Lei, Y., McElroy, M. B., and Hao, J.: Quantifying the uncertainties of a bottom-up emission inventory of anthropogenic atmospheric pollutants in China, Atmos. Chem. Phys., 11, 2295-2308, doi:10.5194/acp-11-2295-2011, 2011.

---

## Author Response (AR2)

**Response to editor:**

Dear Dr. Yugo Kanaya,

We thank you for the constructive comments and suggestions. We have revised the manuscript following the comments as described below. The comments are shown in blue. Our responses are shown in black. The revised texts are shown in italics.

1. The reviewer #1 requested discussion on how the uncertainty in emission inventories propagates into derived contributions. Although the uncertainties are added to Table in supplementary, no discussion is made. Key issues would be how $NO_x$ and anthropogenic VOC concentrations are reproduced by the model simulations. I assume anthropogenic VOC concentrations are measurable from samples for isoprene and monoterpenes. If not, alternative way of evaluation should be proposed.

Thank you for pointing this out. We obtained anthropogenic emissions from the Multi-resolution Emission Inventory for China (Li et al., 2017), which is the most updated bottom-up emission inventory in China. We summarized the uncertainties of the bottom-up emissions in Table S1 in the latest revised manuscript.

As an important model input, emission estimates may largely impact simulated results. For $NO_x$, we evaluated the model performance by comparing with the observed hourly $NO_x$ concentration at urban Xi'an (Table 3 and Figure 3). We found that our model successfully captured the observations of NOx (with a mean bias of -0.4 ppb and NMB=-1%), which suggests no systematic bias in $NO_x$ emissions. Unfortunately, the anthropogenic VOC was not included in our samples as the observations are primarily targeted to biogenic VOC in the Qinling forest. Alternatively, two sensitivity simulations are conducted in this revision (for $15^{th}$-$17^{th}$ August), namely RUN1 (with an increase of anthropogenic VOC emission by 50%) and RUN2 (with a decrease of anthropogenic VOC emission by 33%), to explore the sensitivity of simulated VOC and $O_3$ concentrations to anthropogenic VOC emissions. We found that 50% increases of anthropogenic VOC emission could lead to a 22% increase of urban VOC concentration, while the 33% emission decrease resulted in a 24% decrease of concentration (as shown in the table below). It is worth noting that the concentration of $O_3$ stayed almost the same (because the $O_3$ production regime is $NO_x$-limit). We addressed that the uncertainties of VOC emission obviously affected the VOC concentrations; however, MEIC inventory is the most updated available emission for China so far, and quantifying its uncertainties can be done in future studies (possibly with satellite-based measurement of HCHO (Miller et al., 2008)).

| Species [ ppb ] | Base | RUN1 | RUN2 |
|---|---|---|---|
| **O₃** | **40.7** | **41.9** | **39.8** |
| Ethane | 13.1 | 18.7 | 9.5 |
| C>2 alkanes | 11.2 | 16.1 | 8.9 |
| HCHO | 7.4 | 5.5 | 5.0 |
| Acetaldhyde | 5.4 | 4.7 | 4.2 |
| Toluene | 2.9 | 4.3 | 2.2 |

| | | | |
|---|---|---|---|
| Ethene | 2.8 | 4.2 | 2.3 |
| Organic nitrogen | 2.7 | 1.7 | 1.5 |
| Organic peroxides | 2.1 | 2.0 | 1.9 |
| C>2 alkenes | 2.0 | 3.2 | 1.7 |
| Ketones | 1.6 | 1.6 | 1.2 |
| Xylenes | 1.2 | 1.9 | 0.8 |
| Organic acids | 1.0 | 1.2 | 1.1 |
| **Total VOC** | **53.3** | **64.9** | **40.3** |

We added text in Section 2.3 and 3.3 to discuss about the uncertainties of emission estimates.

*Section 2.3*

*The emission estimates and uncertainties of VOCs, $SO_2$, $NO_x$, $NH_3$, and $PM_{2.5}$ in the domain during the simulation period are summarized in Table S1, and the potential impacts of emission uncertainty on simulation will be discussed in Section 3.3.*

*Section 3.3*

*For NOx, the simulated hourly NOx averaged for the no-raining period was 46.6 ppb, close to the observed 47.0 ppb (NMB=-1%), which suggests no systematic bias in NOx emissions….*

*Unfortunately, the anthropogenic VOC was not included in our samples as the observations are primarily targeted to biogenic VOC in the Qinling forest. Alternatively, two sensitivity simulations are conducted in this revision (for 15th-17th August), namely RUN1 (with an increase of anthropogenic VOC emission by 50%) and RUN2 (with a decrease of anthropogenic VOC emission by 33%), to explore the sensitivity of simulated VOC and O3 concentrations to anthropogenic VOC emissions. We found that 50% increases of anthropogenic VOC emission could lead to a 22% increase of urban VOC concentration, while the 33% emission decrease resulted in a 24% decrease of concentration (Table S2). It is worth noting that the concentration of $O_3$ stayed almost the same (because the $O_3$ production regime is $NO_x$-limit). We addressed that the uncertainties of VOC emission obviously affected the VOC concentrations; however, MEIC inventory is the most updated available emission for China so far, and quantifying its uncertainties can be done in future studies (possibly with satellite-based measurement of HCHO (Miller et al., 2008)).*

2. Similarly, discussion on the potential impact from the isoprene chemical mechanism is still weak. The reviewer #1 requested justification why the new isoprene chemistry mechanism does not impact the results. However such justification is not present, although impact in other studies is cited. The authors mentioned "the impacts of biogenic VOCs on $O_3$ formation may vary in different regions and different seasons." The impact of new isoprene chemistry mechanism could be different too, and therefore case study for Guanzhong Basin is necessary.

Thanks for the comment. We updated the RADM2 mechanism to include the

new isoprene chemistry as follows:

ISOPRENE + OH > ISOPO2

ISOPO2 + NO > ALD + HCHO + $HO_2$ + $NO_2$ + ISOPN

where, ISOPO2 represents the hydroperoxy radical from isoprene, ISOPN represents the organic nitrate.

Many laboratory, filed observations and simulation studies (e.g. von Kuhlmann et al., 2004; Perring et al., 2009; Paulot et al., 2009a; Fisher et al., 2016; Travis et al., 2016) all agreed that tropospheric $O_3$ production was highly sensitive to the ISOPN yield (4% to 15%) and large uncertainties still remained. Horowitz et al. (2007) found a 4% ISOPN yield best captured the alkyl and multifunctional nitrates measured by aircraft, and Hudman et al. (2009) pointed out that unreasonably high ISOPN yield (18%) would let ISOPN be a terminal sink for $NO_x$. Here, we adopted the ISOPN yield to be 4% followed Horowitz et al. (2007) and the rate constant to be $2.7 \times 10^{-12} \times exp(350/T)$ followed MCM3.1. We conducted a 3-day simulation ($15^{th}$ -$17^{th}$ Aug.) to evaluate the potential impacts of this new isoprene chemistry. The simulated results show that near surface $O_3$ concentration decrease by 2-8 ppb over the Qinling Mountains (the region with high isoprene concentration) and increase by 1-5 ppb in the northeast of the Guanzhong basin (impact of the transport of ISOPN). In the urban Xi'an, the averaged $O_3$ concentration decreased by 3 ppb (8%). It could be interesting for our future work to conduct a more detailed analysis about the sensitivity and uncertainties of the ISOPN yield, as well as $NO_x$ recycling.

We added the discussion above in Section 2.2 to state the potential uncertainty.

*We noted some advances in isoprene nitrate chemistry in recent years. Some studies pointed out that isoprene reacts with OH radical to form hydroperoxy radicals (ISOPO2). Subsequently, in the presence of $NO_x$, ISOPO2 reacts with NO leading to the production of hydroxynitrates (ISOPN) by a minor branch, which sequesters $NO_x$ and therefore regulates $O_3$ formation. A number of laboratory, filed observation and simulation studies (e.g. Paulot et al., 2009a, b, 2012; Horowitz et al., 2007; Hudman et al., 2009; Fisher et al., 2016; Travis et al., 2016) highlighted the importance of isoprene nitrate chemistry and all agreed there were still large uncertainties (for example, the estimates of ISOPN yield (4%-15%)). Horowitz et al. (2007) found a 4% ISOPN yield best captured the alkyl and multifunctional nitrates measured by aircraft, and Hudman et al. (2009) pointed out that unreasonably high ISOPN yield (18%) would let ISOPN be a terminal sink for $NO_x$. Back to our study, this isoprene nitrate chemistry was not contained in the standard RADM2 mechanism. To assess the potential uncertainties, we modified the RADM2 mechanism by adding formation pathway of ISOPN from ISOPO2+NO, and run a 3-day ($15^{th}$-$17^{th}$ August) simulation to quantify the impacts in the GZ basin. We adopted the ISOPN yield to be 4% followed Horowitz et al. (2007) and the rate constant to be $2.7 \times 10^{-12} \times exp(350/T)$ followed MCM3.1. A 8% (3 ppb) decrease of the near surface $O_3$ concentration was found averaged for urban Xi'an in August after implementing the updated isoprene nitrate chemistry.*

[Figure]

O₃ change due to isoprene nitrate chemistry

3. The authors mentioned that sampling was conducted between 9:30-16:30 LT. Were the sampling CONTINUED for 7 hours? Or the short-time samplings were done during the period? Clarification is necessary.

We appreciate the editor's concern. Each sampling duration was 30 minutes during 9:30 am to 16:30 pm local time. We modified the text in Section 2.1.1 to make our statement more clear.

*Sampling was conducted between 9:30 am to 16:30 pm local time (each sampling lasted for 30 minutes) to target expected daily maximum isoprene concentrations.*

4. Point 3 of reviewer #2. I also feel impact of aerosols (that the authors claim) on J(NO2), 40-60%, is a bit too large. TUV calculations (http://cprm.acom.ucar.edu/Models/TUV/Interactive_TUV/) to yield 40% lower J(NO2) require AOD=1 and single scattering albedo (SSA) =0.8, or AOD=2 and SSA=0.93. The authors should raise AOD (in addition to PM) and expected SSA during the studied period, to justify that large impact.

Thanks for the suggestion. We calculated the AOD and SSA at 550 nm in the GZ basin during 15[th]-30[rd] August, excluding rainy days. The figure below shows the spatial distribution of the simulated results. The AOD averaged for urban Xi'an is 1.92, and SSA is 0.92. We modified the text in Section 4.1 to describe this issue.

*In the daytime, NO$_2$ photolysis frequency (J(NO$_2$)) is determined by the solar radiation influenced by PM via scattering and absorption. Figure 6 shows the changes of J(NO$_2$) (calculated by model track output photor_no2) with the participation of PM (concentration, 102 µg m$^{-3}$; aerosol optical depth (AOD) at 550 nm, 1.92; single scattering albedo (SSA) at 550 nm, 0.92) averaged for urban Xi'an. J(NO$_2$) was reduced by 40-60%, most significantly in morning and evening rush hours.*

[Figure]

5. Section 4.1. Is the unit for VOCs ppb or ppbC? What are the dominant VOCs for 50 or 30 ppb that the authors discussed?

We appreciate the editor's concern. The unit for VOC is ppb. The dominant VOC in urban Xi'an (the 50 ppb) is ethane, and the domain VOC among the Qinling Mountains (the 30 ppb) is isoprene. We modified the text in Section 4.1 to discuss the issue.

*Figure 4a shows spatial distribution of the simulated VOCs during the no-raining period, overlaid with the simulated wind vectors. The highest concentration (more than 50 ppb, with ethane being the dominant species) was in urban Xi'an and its downwind region (the southwest of urban Xi'an), due to anthropogenic activities. In addition, another high-value area (~30 ppb, with isoprene being the dominant species) was found in the Qinling Mountains, which was probably due to biogenic sources.*

*Manuscript with changes marked*

[revised manuscript text omitted]

---

## Author Response (AR3)

**Response to editor:**

Dear Dr. Yugo Kanaya,

We thank you for the constructive comment. We have revised the manuscript following the comment as described below. The comment is shown in blue. Our response is shown in black. The revised text is shown in italics.

Point regarding isoprene chemistry mechanism needs further justification: Besides sensitivity on nitrate yield, production of radicals from Isoprene RO2 radicals should be discussed. For example, see CheT2 mechanism (ISO2 + N2/O2), in Table 2 of Squire et al. (2015).

Thanks for the comment. The production of radicals from isoprene peroxy radicals (ISOPO2) might be important for secondary pollutants (e.g. $O_3$ and secondary organic aerosol), particularly in isoprene-rich regions. Squire et al. (2015) compared the effects of four different reduced isoprene chemical mechanisms on tropospheric $O_3$. One of the four mechanisms included the formation of hydroperoxy-aldehydes (HPALDs) from ISOPO2 and subsequent rapid release of OH. In addition, Kanaya et al. (2012) evaluated the potential effect of isomerization of ISOPO2 on HO and HO2, in comparison model results with HOx measurements using multiple instruments. They revised the model by adding detailed reactions of isomerization of ISOPO2 and photolysis of HPALDs following Peeters and Müller (2010), and found that the revision could increase OH and HO2 concentrations by 28-38% for daytime. However, Kanaya et al. (2012) also pointed out that the effects of isomerization of ISOPO2 proposed by Peeters and Müller (2010) might be overestimated.

To evaluate the potential impacts of radical from ISOPO2 on tropospheric $O_3$ in the GZ basin, we modified the standard RADM2 mechanism by adding a reduced ISOPO2 isomerization reaction following a very recent study (Li et al., 2018), as follows:

ISOPO2 > 2*HO2+HCHO+0.33*MGLY+0.5*GLYALD+0.25*GLYX+1.5*HACET

with reaction rate coefficient $k = 4.07 \times 10^8 \times exp(-7694/T)$ (see Table R1).

We conducted four sensitivity simulations (15[th]-17[th] Aug.), namely ISO0 (standard RADM2 mechanism), ISO1 (including the above ISOPO2 isomerization reaction), ISO2 (including reactions of isoprene nitrate with the yield of 4%), and ISO3 (including both revisions in ISO1 and ISO2), to explore the effects of the advances in isoprene chemistry. Figure R1 shows spatial distributions of the changes of $O_3$, HOx and dicarbonyls (MGLY+GLYX+GLYALD) due to ISOPO2 isomerization reaction (ISO1-ISO0). We found that ISOPO2 isomerization slightly decreased $O_3$ concentration, less than 0.5 ppb for urban Xi'an (see Table R2), as a result of the HOx increasement. Figure R2 and Table R2 compare the simulated $O_3$ concentration at urban Xi'an from different sensitivity runs. Daily peak $O_3$ concentration decreased by 2.9 ppb (7%) in ISO3 (mix ISOPN production and ISOPO2 isomerization) compared with the ISO0 run. It can be interesting to analyze the budgets of $O_3$ and HOx in future with updated isoprene chemistry in a modified chemistry transport model.

We added the discussion above in Section 2.2 and added Tables R1 and R2 and

Figure R2 in supplementary to state the uncertainty.

[Figure]

**Figure R1.** Concentration changes of $O_3$, $HO_x$ ($HO+HO_2$) and dicarbonyls due to ISOPO2 isomerization reaction (ISO1-ISO0) averaged for $O_3$ peak time (14:00-18:00 local time) in the Guanzhong basin.

[Figure]

**Figure R2.** $O_3$ concentrations changes due to updated isoprene chemistry at Xi'an Jiaotong University for the period of 15[th] to 17[th] August 2011.

**Table R1.** Updated isoprene oxidation chemistry (unit for reaction rate is molecule$^{-1}$ cm$^3$ s$^{-1}$)

| Reactions | Reaction rates |
|---|---|
| ISOPO2+NO >
 ALD+HCHO+HO2+0.96*NO2+0.04*ISOPN | $2.7\times10^{-12}\times exp(350/T)$ |
| ISOPO2 >
 2*HO2+HCHO+0.33*MGLY+0.5*GLYALD+0.25*GLYX
 +1.5*HACET | $4.07\times10^{8}\times exp(-7694/T)$ |

| Species name | Description |
|---|---|
| T | Temperature (K) |

| NO | Nitric oxide |
|---|---|
| NO2 | Nitrogen dioxide |
| HO2 | Hydroperoxy radical |
| ISOPO2 | Hydroperoxy radicals from isoprene oxidation by OH |
| ISOPN | Organic nitrate |
| HCHO | Formaldehyde |
| ALD | Acetaldehyde and higher aldehydes |
| HACET | Hydroxyacetone |
| MGLY | Methylglyoxal |
| GLYALD | Glycolaldehyde |
| GLYX | Glyoxal |

**Table R2.** $O_3$ concentration changes due to updated isoprene chemistry averaged for urban Xi'an

[revised manuscript text omitted]